# Circuits for integrating learned and innate valences in the insect brain

Claire Eschbach[1,2,3]*†, Akira Fushiki[1,4]†, Michael Winding[1,2,3], Bruno Afonso[1], Ingrid V Andrade[1,5], Benjamin T Cocanougher[1,3], Katharina Eichler[1,3], Ruben Gepner[6], Guangwei Si[7,8], Javier Valdes-Aleman[1,3,5], Richard D Fetter[1], Marc Gershow[6,9,10], Gregory SXE Jefferis[2,3], Aravinthan DT Samuel[7,8], James W Truman[1,11], Albert Cardona[1,2,12]*, Marta Zlatic[1,2,3]*

[1]HHMI Janelia Research Campus, Richmond, United Kingdom; [2]Neurobiology Division, MRC Laboratory of Molecular Biology, Cambridge, United Kingdom; [3]Department of Zoology, University of Cambridge, Cambridge, United Kingdom; [4]Department of Neuroscience & Neurology, & Zuckerman Mind Brain Institute, Columbia University, New York, United States; [5]Department of Molecular, Cell and Developmental Biology, University California Los Angeles, Los Angeles, United States; [6]Department of Physics, New York University, New York, United States; [7]Department of Physics, Harvard University, Cambridge, United States; [8]Center for Brain Science, Harvard University, Cambridge, United States; [9]Center for Neural Science, New York University, New York, United States; [10]Neuroscience Institute, New York University, New York, United States; [11]Department of Biology, University of Washington, Seattle, United States; [12]Department of Physiology, Development & Neuroscience, University of Cambridge, Cambridge, United Kingdom

*For correspondence:
ce394@cam.ac.uk (CE);
ac2040@cam.ac.uk (AC);
mzlatic@mrc-lmb.cam.ac.uk (MZ)

†These authors contributed equally to this work

Competing interest: The authors declare that no competing interests exist.

**Abstract** Animal behavior is shaped both by evolution and by individual experience. Parallel brain pathways encode innate and learned valences of cues, but the way in which they are integrated during action-selection is not well understood. We used electron microscopy to comprehensively map with synaptic resolution all neurons downstream of all mushroom body (MB) output neurons (encoding learned valences) and characterized their patterns of interaction with lateral horn (LH) neurons (encoding innate valences) in *Drosophila* larva. The connectome revealed multiple *convergence neuron* types that receive convergent MB and LH inputs. A subset of these receives excitatory input from positive-valence MB and LH pathways and inhibitory input from negative-valence MB pathways. We confirmed functional connectivity from LH and MB pathways and behavioral roles of two of these neurons. These neurons encode integrated odor value and bidirectionally regulate turning. Based on this, we speculate that learning could potentially skew the balance of excitation and inhibition onto these neurons and thereby modulate turning. Together, our study provides insights into the circuits that integrate learned and innate valences to modify behavior.

## Editor's evaluation

This work of Eschbach, Fushiki and colleagues uses optogenetic approaches in the *Drosophila* larval brain to assign behavioral valence to many output neurons of the mushroom body, a brain region canonically associated with assigning learned valence to sensory stimuli. The authors present a comprehensive electron microscopic-level neuroanatomical analysis of the postsynaptic partners of mushroom body putout neurons, identifying substantial convergence from these onto neurons of the lateral horn, a region associated with innate sensory-evoked behavioral responses. Through neurophysiological and behavioral analyses of two classes of these "convergence neurons," they

provide evidence for their role in representing sensory stimuli, evoking behavioral responses, and adapting after learning. Together this work provides an impressive anatomical framework, and initial functional insights, supporting the existence of extensive communication between neural circuits controlling learned and innate behavior.

## Introduction

Selecting appropriate actions in response to sensory stimuli is a major brain function. To achieve this, brains must transform complex representations of sensory stimuli into representations of valences (attractiveness or aversiveness) that can be used to drive actions (*Pearson et al., 2014*). Many sensory stimuli have innate valences, acquired through evolution: some stimuli are innately attractive and others are innately repulsive (*Li and Liberles, 2015*; *Reisenman et al., 2016*). However, to behave adaptively in an ever-changing environment, animals are also able to learn new valences for stimuli. These learned valences can be in conflict with innate ones. For example, repeated association of an innately attractive odor with punishment (e.g. pain or illness) allows a switch from innate attraction to learned aversion of the same odor (*Garcia et al., 1983*; *Tully and Quinn, 1985*; *Pauls et al., 2010*). The innate and learned valences are thought to be encoded in distinct brain areas in both vertebrates (*Li and Liberles, 2015*; *Choi et al., 2011*; *Sosulski et al., 2011*) and invertebrates (*Li and Liberles, 2015*; *Marin et al., 2002*). In mammals, the olfactory projection neurons (mitral cells) send divergent projections to two parallel higher order centers, the olfactory amygdala and the piriform cortex, implicated in innate and learned behaviors, respectively (*Li and Liberles, 2015*; *Choi et al., 2011*; *Sosulski et al., 2011*; *Root et al., 2014*). Likewise, in insects the olfactory projection neurons send divergent projections to the lateral horn (LH) and the mushroom body (MB, [*Marin et al., 2002*; *Wong et al., 2002*; *Gerber and Stocker, 2007*; *Eichler et al., 2017*; *Jeanne et al., 2018*]), implicated in innate and learned behaviors, respectively (*Li and Liberles, 2015*; *Heimbeck et al., 2001*; *McGuire et al., 2001*; *Heisenberg, 2003*; *Turner et al., 2008*; *Ruta et al., 2010*; *Parnas et al., 2013*; *Aso et al., 2014a*; *Aso et al., 2014b*; *Dolan et al., 2019*). Thus, two distinct olfactory structures output valence signals that can be used for an odor response, but the way in which these signals are used to produce a coherent behavioral choice is still an open question. For example, how are conflicting valence signals resolved? Do opposing drives for behavior converge and get integrated, or do they remain in competition (*Pearson et al., 2014*)?

A major obstacle to addressing these questions has been the lack of comprehensive synaptic-resolution maps of the patterns of convergence between neurons that represent innate and learned valences. Another obstacle has been the inability to causally relate specific circuit elements to their function. Here, we were able to overcome these obstacles by using the tractable genetic model system of *Drosophila melanogaster* larva. In this system, we could combine: (i) large-scale electron microscopy reconstruction of neural circuits due to the relatively small size of its brain (*Ohyama et al., 2015*; *Jovanic et al., 2016*); (ii) targeted manipulation of uniquely identified neuron types (*Ohyama et al., 2015*; *Jovanic et al., 2016*; *Tastekin et al., 2018*), (iii) and functional imaging of neural activity.

Previous studies in *Drosophila* have characterized all the components of the MB network and their roles in memory formation and expression. The MB consists of a set of parallel fiber neurons, the Kenyon cells (KCs), that sparsely encode sensory inputs coming from olfactory and other projection neurons (PNs [*Gerber and Stocker, 2007*; *Honegger et al., 2011*; *Papadopoulou et al., 2011*; *Lin et al., 2014*; *Hige et al., 2015b*]) KC axons are tiled into distinct compartments by terminals of modulatory neurons, mainly dopaminergic (DANs [*Eichler et al., 2017*; *Aso et al., 2014a*; *Mao and Davis, 2009*; *Takemura et al., 2017*]). DANs carry information about positive and negative reinforcement and provide teaching signals for memory formation (*Aso et al., 2014a*; *Schroll et al., 2006*; *Liu et al., 2012*; *Aso and Rubin, 2016*; *Eschbach et al., 2020*). In each compartment, DANs synapse onto KCs and onto the dendrites of compartment-specific MB output neurons (MBONs [*Eichler et al., 2017*; *Takemura et al., 2017*]). In the adult, individual MBONs have been shown to promote approach or avoidance and encode positive or negative valence, respectively (*Aso et al., 2014b*; *Owald and Waddell, 2015*; *Bouzaiane et al., 2015*). Pairing of an odor with a DAN has also been shown to selectively depress the conditioned-odor drive to MBONs in that compartment (*Séjourné et al., 2011*; *Hige et al., 2015a*). Prior to learning, positive- and negative-valence MBONs are thought to receive a similar odor drive. Aversive and appetitive learning depresses the odor drive to positive- and

negative-valence MBONs, respectively (*Aso et al., 2014b*; *Owald and Waddell, 2015*). Learned valence of stimuli is therefore thought to be encoded as a skew in the activity of the population of MBONs. However, despite recent progress (*Dolan et al., 2019*; *Dolan et al., 2018*; *Scaplen et al., 2020*; *Bates et al., 2020*; *Schlegel et al., 2021*), the way in which the learned valence is read out by the downstream networks and used to select actions and the way in which innate and learned valences are integrated is still poorly understood.

Here, we investigate the circuit mechanisms by which innate and learned valences interact in *Drosophila* larvae to regulate turning response to odors. We determined which larval MBONs encode positive valence and repress turning and which ones encode negative valence and promote turning. We exhaustively reconstructed all neurons postsynaptic to all MBONs, as well as all LHNs in an innately attractive pathway. These reconstructions revealed the structural patterns of convergence between the brain areas that encode innate and learned valences. We found that (i) some MBONs directly synapse onto LHNs, (ii) some LHNs directly synapse onto some MBONs, and (iii) some MBONs and LHNs converge onto distinct subtypes of 'convergence neurons (CNs)'. One CN subtype is activated by attractive odors via the LH pathway and its activation represses turning (to allow approach). These CNs also receive excitatory and inhibitory inputs from MBONs that encode opposite valence. As we showed for one of these CNs, the balance of mixed MB inputs can be modified by learning to modulate odor-evoked responses and thereby turning. These CNs may therefore integrate learned and innate valences and regulate turning based on the integrated value. Our study provides mechanistic insight into how conflict between opposing valences can be resolved by a population of integrative neurons. Furthermore, the connectome of the circuits for integrating learned and innate valences in the larval brain (available at https://l1em.catmaid.virtualflybrain.org/?pid=1) provides an essential basis for further modelling and functional studies of value computation and action selection.

## Results
### Associative learning modulates turning response to odors

*Drosophila* larvae are innately attracted by most volatile molecules (*Fishilevich and Vosshall, 2005*; *Kreher et al., 2008*; *Mathew et al., 2013*), and repelled by a few, such as $CO_2$ (*Gershow et al., 2012*). Larvae navigate gradients of innately attractive or repulsive odors via klinotaxis (*Gershow et al., 2012*; *Gomez-Marin and Louis, 2012*; *Schulze et al., 2015*; *Gepner et al., 2015*), by modulating turning and crawling in response to changes in odor concentration. This navigational strategy involves turning every time the animal is moving in the bad direction, and not turning or actively repressing turning, every time the animal is moving in the good direction. By doing that, the animal will, on average, keep moving in the good direction and correct its course when moving in the bad direction. In the context of navigating odor gradients, this strategy involves turning when moving toward an aversive odor source (bad direction), or when moving away from an appetitive odor source (also bad direction because it takes the animal away from the source of appetitive odor). Thus, larvae approach innately attractive odors as follows: (i) When crawling towards the attractive odor source they sense an increase in odor concentration so they repress turning (and promote crawling); (ii) when crawling away from the attractive odor source they sense a decrease in odor concentration so they promote turning (and repress crawling). They avoid innately aversive odors by doing the opposite. Associative odor learning has been shown to modify the turning response to odor (*Schleyer et al., 2015*; *Paisios et al., 2017*). Here, we report similar findings using optogenetic punishment. We compared groups of *ca.* 30 larvae that received an innately attractive odor, ethyl acetate (volume dilution of $10^4$ fold in water) (*Kreher et al., 2008*; *Gershow et al., 2012*), paired with the activation of the nociceptive Basin interneurons (*Ohyama et al., 2015*) to larvae that received unpaired presentation of the two stimuli (*Figure 1a–c*). As expected, we found that after the innately attractive odor was paired with optogenetic punishment, on average, larvae no longer approached it, and instead avoided it (*Figure 1b*). To account for this behavioral change, we computed a navigation index as the overall distance crawled in the direction up the stimulus gradient divided by the total distance crawled. Following unpaired odor presentation this index reached +0.1, that is similar to previous studies looking at naive larvae navigating toward ethyl acetate, at this dilution (*Gershow et al., 2012*). Following odor presentation paired with Basin activation, the index reached –0.1, that is similar to larvae navigating away from $CO_2$ (*Gershow et al., 2012*). This involved a change in the coupling

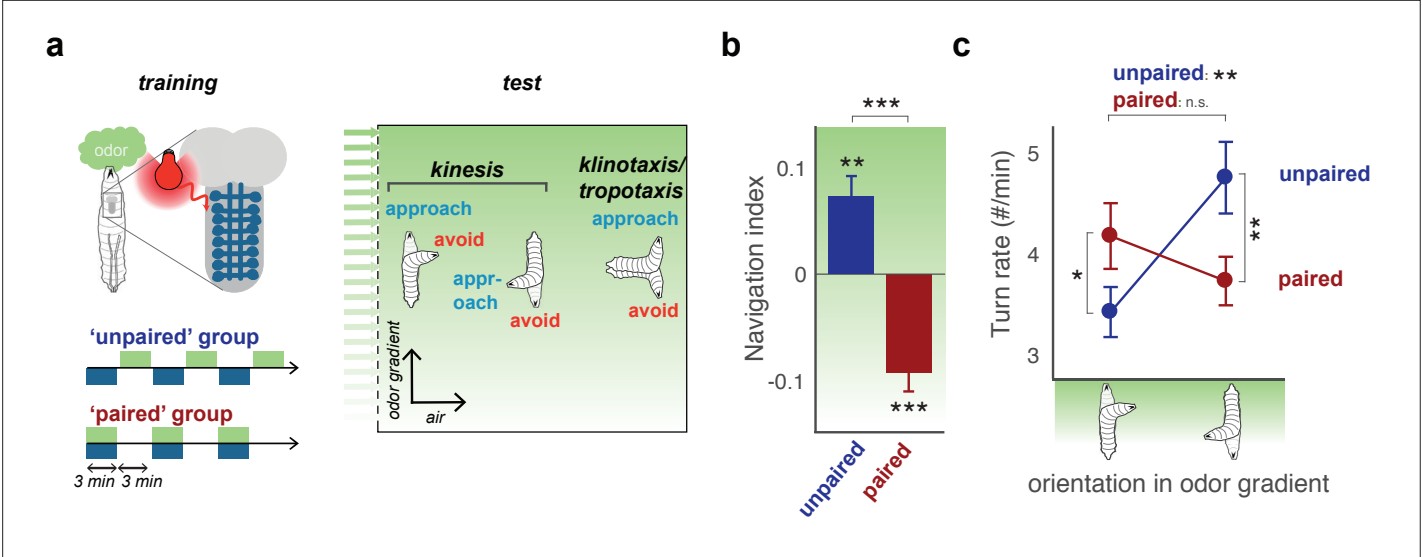

**Figure 1.** Aversive learning modulates turning to switch odor approach to avoidance. (**a**) The behavior of larvae in a linear gradient (**Gershow et al., 2012**) of an innately attractive odor, ethyl acetate, (EA) is recorded after odor presentation is intercalated ('unpaired' training protocol) or coincident ('paired' training protocol) with fictive punishment (optogenetic activation of nociceptive basin neurons [**Ohyama et al., 2015**]). Larvae navigate via kinesis, modulating turn rate over time and in response to different conditions, and via klinotaxis/tropotaxis, choosing turn side (**Gershow et al., 2012**; **Gomez-Marin and Louis, 2012**; **Schulze et al., 2015**; **Gepner et al., 2015**). Here, we record turn rate as a function of larval orientation in the odor gradient (**c**). (**b**) Navigation index following paired and unpaired protocols (computed by dividing the mean velocity in the direction of the gradient by the mean crawling speed). The positive index after the unpaired protocol indicates that larvae approach the odor, the negative index after the paired protocol indicates they avoid it. (**c**) Turn rate as a function of larval orientation in the gradient. Unpaired group larvae approach the odor by turning less in response to an increase in odor concentration (when crawling up the gradient, toward the odor source), and more in response to a decrease in odor concentration (when crawling down the gradient, away from the odor source). Paired group larvae avoid the odor by turning more in response to an increase in the odor concentration (when crawling up the gradient), and by turning less in response to a decrease in odor concentration (when crawling down the gradient). Values are mean s.e.m. *: p < 0.05, **: p < 0.01 in a Welch Z-test, N = 10 repeats.

between odor intensity change and turn probability: the paired group turned more when crawling up the odor gradient, than down the odor gradient, whereas the unpaired group larvae did the opposite (**Figure 1c**). The rest of this article will focus on investigating the neural basis of altering the turning response to odor following aversive learning.

## Identification of positive- and negative-valence MBONs that repress and promote turning

Since olfactory learning has been found to modify the strength of KC-to-MBON synapses in the adult (**Séjourné et al., 2011**; **Hige et al., 2015a**), we first investigated whether and how individual MBONs influence turning. Based on studies in the adult (**Aso et al., 2014b**), we hypothesized that some MBONs encode positive valence, while others encode negative valence. Since positive valence neurons are activated by attractive odors, an increase in their activity is expected to have the same effect on behavior as an increase in the concentration of an attractive odor (see above). Conversely, negative valence neurons are activated by aversive odors, so an increase in their activity is expected to have the same effect on behavior as an increase in the concentration of an aversive odor (see above). A neuron whose activation signals positive valence is therefore expected to suppress turning in response to an increase in its activity, and promote turning in response to a decrease in its activity. Conversely, a neuron whose activation signals negative valence (avoidance) is expected to promote turning in response to an increase in its activity. We therefore expected to find MBONs that either promote or suppress turning in response to an increase in their activity.

We generated Split-GAL4 lines (**Pfeiffer et al., 2010**) that allowed us to drive expression of the red-shifted channelrhodopsin CsChrimson (**Klapoetke et al., 2014**) in a single or in a couple of indistinguishable MBONs per brain hemisphere (**Figure 2—figure supplement 1**). We then monitored behavioral responses to optogenetic activation of individual MBON types (**Figure 2a–g**). Activating

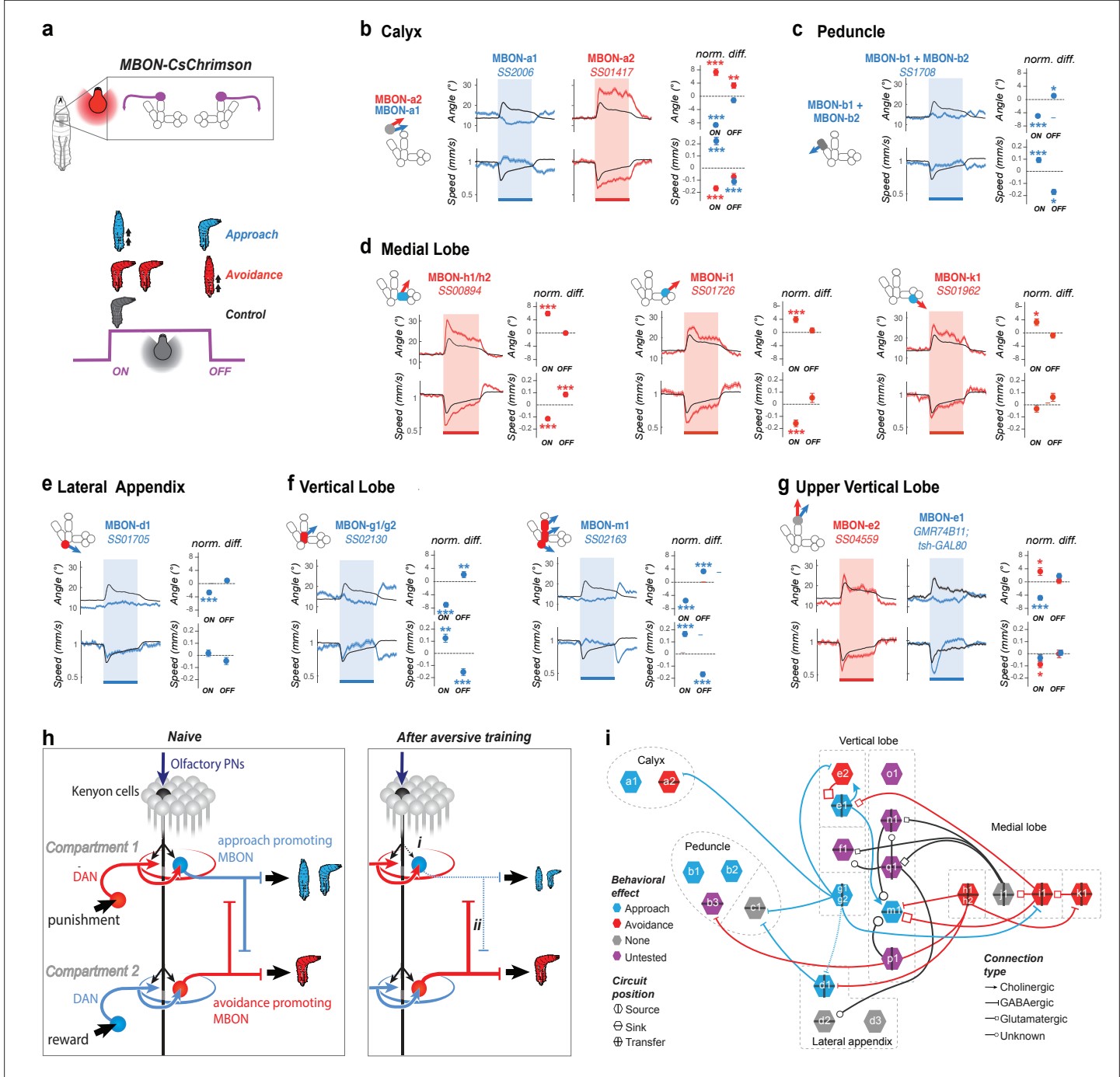

**Figure 2.** Positive- and negative-valence MBONs repress and promote turning, respectively. (**a**) We recorded the behavior of larvae expressing CsChrimson in one or two pairs of MBONs and classified their responses to a 15 s red light stimulation as approach-like (*blue larva*), avoidance-like (*red larva*), or neutral by comparing them to controls. Approach-like responses are characterized by significantly decreased turning and/or increased crawling speed in response to an increase (at the onset of optogenetic activation), and/or increased turning and decreased crawling in response to a decrease in MBON activity (at the offset of optogenetic activation). Avoidance-like responses are characterized by the reverse responses. We classified MBONs whose activation suppresses turning and promotes approach as positive-valence MBONs (*blue* in **b–i**), and those whose activation promotes turning and promoted avoidance as negative-valence MBONs (*red* in **b–i**). (**b–g**) Behavioral response to optogenetic activation of MBONs (GAL4 lines indicated in italic; see *Figure 2—figure supplement 1* for expression patterns). Turn angle (deg) is the absolute value of the distance from the least-squares line fit of the posterior 2/3 of the animal's spine points to the point in the anterior 1/5 of the animal's spine most distant from that line. *Left*, schematics depicting compartments innervated by MBONs, colored according to memory induced when odor is paired with DAN activation in that compartment: appetitive (*blue*), aversive (*red*), or unknown (*gray*, *Eschbach et al., 2020*). *Middle*, time series of mean (+/− s.e.m) turn angle (*top*)

*Figure 2 continued on next page*

*Figure 2 continued*

and crawling speed (normalized to baseline prior to stimulation, *bottom*). Shading indicates the period of optogenetic activation. *Right*, the difference between the experimental (*red* or *blue* dots with error bars) and control (*dotted line at 0*) turn angle (*top*) and crawling speed (*bottom*) averaged over a time window after the onset (0–5 s, normalized to baseline before light) and offset (2–7 s. after light off, normalized to baseline during light) of optogenetic stimulation. Note that the control animals (*black curve* in **b–g**) display a slightly aversive response to the onset of red light used for optogenetic activation. The control is the empty GAL4 line *y w;attP40;attP2* crossed to *UAS-CsChrimson* (Nexp = 343, Nlarvae >10,000) for all lines except for MBON-e1 (**g**) for which the control line is *yw;;attP2* crossed to *UAS-CsChrimson; tsh-GAL80*. Plots are mean +/− s.e.m. *: p < 0.05, **: p < 0.01, ***: p < 0.001 (Welch's Z test). (**b**) Activating the two calyx-MBONs induced opposite responses: approach and avoidance, for MBON-a1 (Nexp = 7, Nlarvae = 250) and MBON-a2 (Nexp = 7, Nlarvae = 280), respectively. (**c**) Activating peduncle-MBONs, MBON-b1/b2 together (Nexp = 8, Nlarvae = 340) induced approach. MBON-c1 activation had no significant effect (*Figure 2—figure supplement 2*). (**d**) Activating medial lobe MBONs induced avoidance: MBON-h1/h2 (Nexp = 10, Nlarvae = 450), MBON-i1 (Nexp = 6, Nlarvae = 240), and MBON-k1 (Nexp = 6, Nlarvae = 210). MBON-j1 activation had no significant effect (*Figure 2—figure supplement 2*). (**e**) Activating the lateral appendix-MBON-d1 (Nexp = 9, Nlarvae = 250) induced approach. MBON-d2 and -d3 activation did not have a significant effect (*Figure 2—figure supplement 2*). (**f**) Activating the vertical lobe-MBONs induced approach: MBON-g1/g2 (Nexp = 6, Nlarvae = 210) and MBON-m1 (Nexp = 9, Nlarvae = 450). (**g**) Activating the two MBONs in the tip of vertical lobe had opposite effects: MBON-e2 (Nexp = 5, Nlarvae = 140) and -e1 (Nexp = 6, Nlarvae = 250, using *tsh-GAL80* to eliminate nerve cord expression) induced avoidance- and approach-like responses, respectively. Note that most negative-valence MBONs innervate appetitive-memory com- partments (3/5), and the remainder innervate compartment with unknown roles. By contrast, most positive-valence MBONs (4/6) innervate aversive-memory compartment, and the remainder innervate compartments with unknown roles. (**h**) Schematic showing a naive state (*left*) and two main mechanisms (*right*) potentially enabling the MB network to switch from encoding positive or neutral to negative valence after aversive learning. *Left*, In a naive state KC-connections to positive- and negative-valence MBONs is similar. *Right*, (i) aversive learning depresses the synapse between the conditioned odor-KCs and positive-valence MBONs (*Hige et al., 2015b*), skewing the balance towards negative-valence MBONs. (ii) Decreased conditioned odor drive to inhibitory positive-valence MBONs can disinhibit the negative-valence MBONs. (**i**) Synaptic-resolution circuit diagram from *Eichler et al., 2017* overlaid with MBON neurotransmitter profiles also from *Eichler et al., 2017* and valence (**b–g**), reveals lateral inhibition between MBONs that encode opposite valence. *Blue rim*, positive-valence; *red rim*, negative-valence; *grey rim*, no behavioral effect; *purple*; not tested. *Arrows*, excitatory cholinergic; *bars*, inhibitory GABAergic; *squares*, glutamatergic, likely inhibitory (*Liu and Wilson, 2013*) connections; *circles*, unknown neurotransmitter. *Vertical* and *horizontal bars*, source (i.e. emitting projections), sink (i.e. receiving projections), or transfer MBONs (i.e. emitting and receiving projections).

The online version of this article includes the following figure supplement(s) for figure 2:

**Source data 1.** Value angles showing turn responses of animals during activation of MBONs.

**Figure supplement 1.** Expression patterns of Split-GAL4 lines.

**Figure supplement 2.** Some MBONs do not promote visible response when optogenetically activated.

**Figure supplement 3.** Reverse-correlation analysis of turn behavior with optogenetics.

some MBONs had no effect on behavior (*Figure 2—figure supplement 2*). Most MBONs that we tested fell into one of two opposing categories in terms of their effect on behavior.

Activating some MBONs repressed turning and promoted crawling, compared to controls (*Figure 2b–c and e–g*). In most cases, the decrease in turn is also associated with an increase in speed (except for MBON-d1 and MBON-e1, *Figure 2e and g*). Additionally, some of these MBONs (-a1, -g1/g2 and -m1) promote turning and/or repress crawling in response to a decrease in their activity, that is at the offset of optogenetic activation (*Figure 2b–c , and f*). We group these diverse profiles as positive-valence MBONs, since their activation represses turning. If they were activated by an increase in odor concentration (as would occur when the animal is crawling towards an odor source) they would repress turning, allowing the animal to approach the odor. Also, if the activity of some of these neurons is decreased (as would occur if the animal is crawling away from an odor source) they would promote turning. Thus, while all positive-valence MBONs repressed turning when activated they differed in terms of their effect on speed and in terms of the response to a decrease in their activity. This raises the possibility that different types of appetitive memories could be differentially expressed (e.g. with different speeds of crawling or efficacy of navigation).

Activation of other MBONs promotes turning, compared to controls (*Figure 2b, d and g*), and typi- cally also represses crawling speed (except for MBON-k1, *Figure 2d*). We classify these as negative-valence MBONs. If these neurons were activated by an increase in odor concentration (as would occur when the animal is crawling towards an odor source) they would promote turning which would result in odor avoidance. Notably, different negative-valence MBONs have different effects on the duration of the turn response: the response to some MBONs adapts more quickly with turning increased only shortly after the activation onset (e.g. MBONh1/h2, MBON-i1), while others induce a more sustained turn increase that lasts the duration (whole 15 s) of the light exposure (e.g. MBON-a2, MBON-k1). Some of these MBONs (-h1/h2 and -i1) also promoted crawling in response to a decrease in their

activity. Thus, while all negative-valence MBONs promote turning when activated, they differ in terms of the offset response and the duration of the turning response. This suggests that the MB could potentially allow different types of aversive memories to be differentially expressed (with varying durations of turns, speeds of crawling or efficacies of navigational strategies).

In our experiments we observed a turn response to the red light used for optogenetic activation in control animals. We wanted to rule out the possibility that MBONs only modulate the light-evoke startle and confirm they could modulate turning even in the absence of a light-evoked turn. For a subset of positive- and negative-valence MBONs, we therefore repeated the optogenetic manipulation experiments in a different way, by constantly exposing larvae to dim blue light of constant intensity and bright red light (for CsChrimson activation) of randomly varying intensity (*Figure 2—figure supplement 3a-c*). Under these conditions, control larvae habituated to light and did not show light-induced startle responses. Reverse-correlation analysis (*Gepner et al., 2015*; *Hernandez-Nunez et al., 2015*; *Gepner et al., 2018*) between turning and red light intensity revealed that larvae with CsChrimson in negative-valence MBONs turn following an increase in red light intensity (i.e. an increase in MBON activity), and larvae with CsChrimson in positive-valence MBONs turn following a decrease in light intensity (i.e. a decrease in MBON activity, *Figure 2—figure supplement 3a-b*). Also, for negative-valence MBONs, the greater the change in light intensity during a turn, the higher the probability of rejecting that direction and performing another turn. Conversely, for positive-valence MBONs, the greater the change in light intensity during a turn, the higher the probability of accepting that turn direction (*Figure 2—figure supplement 3a and c*). This indicates that activity in MBONs impacts multiple behavioral components involved in avoidance or approach of a sensory stimulus in a way that is consistent with their valence.

We found that most positive-valence MBONs innervate compartments implicated in aversive memory formation and receive synaptic input from DANs whose activation (paired with odor) induces aversive memory (*Eschbach et al., 2020*; *Figure 2e–f , and h*). Conversely, most negative-valence MBONs innervate compartments implicated in appetitive memory formation and receive synaptic input from DANs whose activation (paired with odor) induces appetitive memory (*Eschbach et al., 2020*; *Rohwedder et al., 2016*; *Saumweber et al., 2011*; *Figure 2d and h*). Curiously, some MB compartments of unknown function (Upper Vertical Lobe and Calyx) are innervated by two distinct MBONs that had opposite effects on turning behavior (*Figure 2b and g*).

Overall our findings are consistent with mechanisms described in the adult *Drosophila* (*Aso et al., 2014a*; *Owald and Waddell, 2015*): the formation of an aversive olfactory memory reduces the conditioned odor drive to positive-valence MBONs that repress turning, and vice versa, for appetitive memory (*Figure 2h*).

## Lateral inhibition between MBONs of opposite valence

To begin to understand how the activity of the entire population of MBONs is used to control learned odor responses we first analyzed direct interactions between MBONs of opposite valence that have opposite effects on turning behavior. We have recently mapped the synaptic-resolution connectivity between all MBONs in a first instar larval brain and identified their neurotransmitter expression (*Eichler et al., 2017*). We therefore combined the behavioral effects of MBON activation with this information (*Figure 2i*). We noted lateral inhibition between some MBONs that promote opposite behaviors: 6 instances of an avoidance-promoting MBON projecting inhibitory synapses (i.e. GABAergic or glutamatergic [*Liu and Wilson, 2013*]) onto an approach-promoting MBON, and hree instances of an approach-promoting MBON projecting inhibitory synapses onto an avoidance-promoting MBON.

We postulate that for these neurons disinhibition could enhance the contrast in odor drive to approach- and avoidance-promoting MBONs (*Figure 2h*). In the example of an aversive olfactory memory, the depressed odor drive onto approach-promoting MBONs would be accompanied by reduced inhibition of avoidance-promoting MBONs. Such disinhibition has been shown following aversive learning in the adult (*Owald et al., 2015*). In the larva, we find that several approach- and avoidance-promoting MBONs are targets of lateral inhibition by MBONs of opposite valence, suggesting that existence of lateral inhibition between some MBONs that promote opposite actions may be a general principle of MB organization (*Figure 2h and i*). However, considering that this motif is not present between all MBONs driving approach or avoidance, additional mechanisms would be

needed to account for a switch between approach and avoidance. We therefore investigated if interactions between MBONs could be found one synapse downstream of the MBONs.

## Comprehensive EM reconstruction of all neurons downstream of all MBONs reveals candidate neurons for comparing odor drive to MBONs of opposite valence

To investigate how downstream circuits could compare the odor drive to positive- and negative-valence MBONs and read out the learned odor valence, we reconstructed all neurons postsynaptic to all 24 MBONs in both the right and left hemispheres (*Figure 3a–b*, *Figure 3—figure supplement Figure 3—figure supplements 1 and 2a-e* and 3a-d, *Supplementary file 1*). We identified 167 left and right homologous pairs of neurons that were strongly and reliably connected to MBONs (see Materials and methods for definition of strong and reliable, *Figure 3b*). We named these neurons MB second-order output neurons (MB2ONs).

40/167 MB2ONs synapse directly onto MB modulatory neurons, and had been reconstructed as part of our investigation of modulatory neuron inputs (*Eschbach et al., 2020*). We have previously named these neurons, feedback neurons (FBNs). Another 58/167 neurons provide indirect two-step feedback to modulatory neurons by synapsing onto a pre-modulatory neuron (*Supplementary file 1*).

We observed both divergence and convergence of MBON inputs at the downstream layer. Many MB2ONs (101/167) receive inputs from only one MBON (*Figure 3—figure supplements 1 and 2a*) and each MBON synapses onto multiple MB2ONs (*Figure 3—figure supplements 1 and 2b*, *Supplementary file 1*). Consequently, each MBON projects to a unique combination of MB2ONs (*Figure 3—figure supplements 1 and 2c*, *Supplementary file 1*). Nevertheless, we observed a large population of 66 MB2ON types that received convergent input from multiple MBONs (*Figure 3—figure supplements 1 and 2a*). Some integrate input from MBONs of the same valence (13/66), but many more integrate input from MBONs of opposite valence (27/66, *Figure 3—figure supplement 1*). Interestingly, many of these (18/27) appear to receive excitatory connections from MBONs of one valence and inhibitory connections from MBONs of opposite valence (*Figure 3—figure supplement 1*). These MB2ONs could compare the odor drive to MBONs of opposite valence and thereby compute the overall learned valence of an odor based on memory traces from multiple compartments.

## EM reconstruction of LH neurons reveals how LH and MB pathways converge

Next, we asked how the learned valence signals from the MB are integrated with the innate valence signals from the LH, for example, during aversive learning of an innately attractive odor. We therefore sought to: (1) Identify all the LHNs downstream of olfactory projection neurons (PNs) with strongest response to an innately attractive odor, ethyl acetate; (2) Confirm that this LHN pathway is sufficient to mediate innate olfactory behavior even in the absence of the MB and that it encodes positive valence (i.e. that its activation represses turning); (3) Determine the patterns of synaptic connections between these LHNs, MBONs, and MB2ONs.

Olfactory receptor neurons (ORNs), ORN42a and ORN42b (*Kreher et al., 2008*; *Kreher et al., 2005*) show the strongest response to the innately attractive odor, ethyl acetate. An increase in the activity of ORN42a or ORN42b in untrained animals decreases turning, while a decrease in their activity increases turning (*Schulze et al., 2015*; *Gepner et al., 2015*; *Hernandez-Nunez et al., 2015*), indicating they promote approach of innately attractive odors. ORN42a and ORN42b synapse onto PN42a and PN42b, which bifurcate, sending projections to both the MB and the LH. While all of the olfactory PNs and KCs were recently reconstructed in an EM volume of a larval nervous system (*Berck et al., 2016*), the neurons downstream of PN42a and PN42b, other than KCs, were previously unknown. We therefore reconstructed all neurons downstream of PN42a and PN42b in the same EM volume and identified 22 pairs of LHNs (LHNs, *Figure 3a–f*, *Figure 3—figure supplement 3a-b*, *Supplementary file 1*).

Second, we wanted to confirm that the reconstructed LHN pathway is sufficient to mediate innate olfactory behavior and that it encodes positive innate valence (i.e. that its activation represses turning). Since we do not yet have suitable GAL4 lines to selectively target these LHNs, we selectively activated this LHN pathway indirectly, by activating the upstream ORN42b, while at the same time inactivating the MB pathway. Since PN42b axons synapse only onto KCs and onto these LHNs, when KCs are

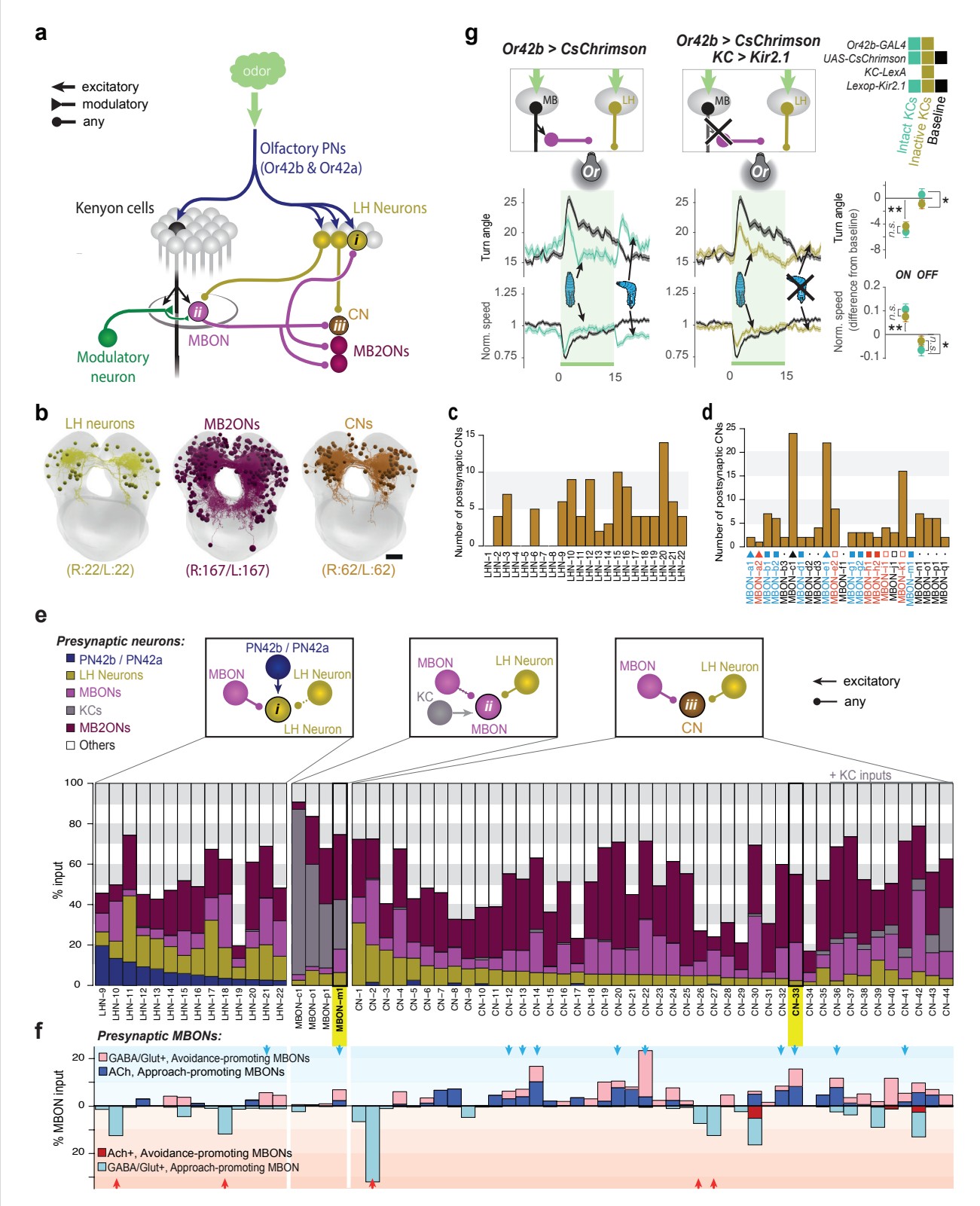

**Figure 3.** Patterns of convergence between innate and learnt olfactory pathways. (**a**) Schematic of interactions between the LH and MB. All neurons (167 pairs) postsynaptic to all MBONs (MB2ONs, *dark purple*) and all neurons (22 pairs) postsynaptic to two olfactory PNs in an innately attractive LH pathway (LHNs, *dark yellow*, ***Figure 3—figure supplement 1***). This revealed three types of convergence: (i) some LHNs receive direct MBON input, (ii) some MBONs receive direct LHN input, and (iii) many MB2ONs receive direct MBON and LHN input. Collectively we call neurons that receive convergent

*Figure 3 continued on next page*

*Figure 3 continued*

MBON and LHN input convergence neurons (CNs). Some LHNs that receive direct MBON input also receive reliable input from other LHNs, and some MBONs (MBON-m1) that receive direct LHN input also receive reliable input from other MBONs(e), and are therefore also CNs (LHN-CNs and MBON-CNs), but we continue to call them LHNs and MBONs for brevity. (**b**) Projections of the reconstructed brain neurons. (**c**) Number of CNs downstream of each LHN. Some target more CNs than others. (**d**) Number of CNs downstream of each MBON. Some target more CNs than others. (**e**) All identified neurons receiving convergent LH and MB inputs. Top schematics show the three different types of convergent interactions described in (**a**). The bar graphs show the fraction of inputs each neuron receives from different types of neurons. Some CNs receive direct KC input, but were not classified as MBONs because connections from individual KCs were weak ( < 3 synapses) or asymmetric ( ≥ 3 KC connection(s) only present in one hemilateral partner) (CN-35,–36, −37,–38, −39,–40, –42), KC inputs were axo-axonic (CN-43), or do not receive any modulatory neuron input (CN-44, *Eichler et al., 2017*). Some CNs receive asymmetric or subthreshold PN input (*e.g.* CN-5) and are therefore not considered LHNs. f. Some CNs were predicted to encode positive (*blue arrows*) or negative (*red arrows*) value based on MBON inputs. Figure shows fraction of input each neuron receives from inhibitory (*light blue, bottom*) or excitatory (*dark blue, top*) positive-valence MBONs (respectively 6 and 2 MBONs form these categories), and inhibitory (*light red, top*) or excitatory (*dark red, bottom*) negative-valence MBONs (respectively 5 and 1 MBONs form these categories). CNs receiving excitatory and inhibitory inputs from positive- and negative-valence MBON(s), respectively, or inhibitory inputs from multiple negative-valence MBONs could encode positive valence (*blue arrows*). CNs integrating inhibitory inputs from multiple positive-valence MBONs could encode negative valence (*red arrows*). Only neurons with at least 5% of input from the appropriate MBONs were included in the predictions (*blue arrows, red arrows*). g. Increase in ORN42b activity (as when the animal crawls towards an odor source) represses turning via the LH pathway, indicating this LH pathway signals positive valence. Plots and quantification as in *Figure 2b–f*. *Left*, In untrained larvae, onset and offset of optogenetic activation of ORN42b reduced and increased turning, respectively (*cyan*, N = 250), compared to controls (*black*, N = 350). *Middle*, Silencing KCs with Kir2.1 (*dark yellow*, N = 300) did not alter the onset response but abolished the offset response. *Right*, quantification. *: p < 0.05, **: p < 0.01 (Welch's Z test).

The online version of this article includes the following figure supplement(s) for figure 3:

**Source data 1.** Value angles showing turn responses of animals during activation of Or42b with or without functional MB.

**Figure supplement 1.** Connectivity matrix.

**Figure supplement 2.** Analysis of M2BONs inputs.

**Figure supplement 3.** Circuit diagram of MBONs and LHNs.

**Figure supplement 4.** Functioning Kenyon Cells are required for normal approach behavior of an odor.

**Figure supplement 5.** AnalysiCof CNs inputs.

**Figure supplement 6.** Clustering of CNs based on their different kinds of inputs.

---

inactivated, the only pathway downstream of ORN42b that can support behavior is the LHN pathway. We therefore compared turning response to optogenetic activation of ORN42b in larvae with silenced (using the potassium channel Kir2.1 [*Baines et al., 2001*]) or intact KCs (*Figure 3g*). In both groups, we observed a comparable and significant decrease in turning in response to an increase in ORN42b activity (*Figure 3g*). This confirms that the LHN pathway downstream of PN42b suppresses turning when activated by ORN42b, even in the absence of a functional MB pathway, and encodes positive valence (*Figure 3g*).

Interestingly, compared to animals with intact KCs, larvae with silenced KCs responded differently to a reduction in ORN42b activity at the offset of optogenetic activation (*Figure 3g*). This suggests that the MB contributes to some aspects of the innate odor response, specifically to turning following a decrease in odor concentration (as would occur when the animal is crawling away from an innately attractive odor source). Consistent with this idea, navigation in a gradient of ethyl acetate (*Figure 3—figure supplement 4*) was less efficient when KCs were silenced. Notably, a defect in innate odor attraction has also been observed in adult *Drosophila* with silenced KCs (*Wang et al., 2003*), at low odor concentration.

Next, we analyzed the anatomical patterns of interaction between LHNs, MBONs, and MB2ONs (*Figure 3a–f* and *Figure 3—figure supplement 3b*). Our EM reconstruction revealed direct connections from some MBONs onto 14 LHNs, similar to recent findings in the adult *Drosophila* (*Dolan et al., 2018*; *Figure 3a and e*, and *Figure 3—figure supplement 3b i*). We also observed direct connections from some LHNs onto four MBONs (*Figure 3a and e*, and *Figure 3—figure supplement 3b,ii-iii*) and the convergence of both LHNs and MBONs onto 44 MB2ONs (*Figure 3a and e*). Collectively, we call the neurons that receive LHN and MBON input, 'Convergence Neurons' (CNs). Most LHNs that receive direct MBON input also receive reliable input from other LHNs, and some MBONs (e.g. MBON-m1) that receive direct LHN input also receive reliable input from other MBONs, and are therefore also CNs (LHN-CNs and MBON-CNs), but we continue to call them LHNs and MBONs for brevity (*Figure 3e*). Distinct CNs receive distinct patterns of inputs from LHNs, MBONs, and other MB2ONs (*Figure 3—figure*

*supplement 5* and *Figure 3—figure supplement 6*, *Supplementary file 1*). Interestingly, many CNs are also feedback neurons (18) and synapse directly onto modulatory neurons (*Eschbach et al., 2020*) (we name these CN/FBN, but refer to them as CN here, for brevity, see also *Supplementary file 1*).

Based on their input from positive- and negative-valence MBONs we postulated that CNs could fall into distinct functional classes (*Figure 3f*). We predicted the signs of connections (inhibitory or excitatory) made by MBONs based on their known neurotransmitter expression (*Eichler et al., 2017*). Because we have not yet generated GAL4 lines for targeting LHNs in the larva, we could not determine their neurotransmitter identity. Some CNs receive a significant fraction of input (more than 5%) from inhibitory positive-valence MBONs and others from inhibitory negative-valence MBONs, suggesting their activation potentially encodes negative and positive valence, respectively. A prominent CN subtype receives input both from excitatory positive-valence MBONs and inhibitory negative-valence MBONs (N = 10). Activating the candidatepositive valence CNs could potentially suppress turning like the MBONs that excite them. In the following sections we test the predictions about the roles of two of these putative 'positive-valence' CNs, MBON-m1 and CN-33, by analyzing their functional connectivity from LH and MB and effects on behavior.

## MBON-M1 is excited by innately attractive odors via the LH pathway

If activating CNs that receive excitatory input from positive-valence MBONs indeed signals positive valence, we would expect that they are activated by innately attractive odors via the LH pathway. We wanted to test whether this is indeed the case by imaging calcium responses of these CNs to an innately attractive odor, in the presence, or absence, of a functional MB pathway. To do this, we needed the ability to selectively target GCaMP expression to these neurons. We have not yet generated Split-GAL4 lines for most of the newly identified CNs and LHNs, but we have generated Split-GAL4 lines for many MBONs (*Figure 2—figure supplement 1*). We therefore focus our initial investigation on the CN-MBON-m1 that integrates direct synaptic inputs from LHNs downstream of ORN42b PNs, from KCs, and from other MBONs (*Figure 4a*). Specifically, MBON-m1 receives cholinergic (excitatory) input from positive-valence MBON-e1 that repress turning and GABAergic (inhibitory) and glutamatergic (likely inhibitory [*Liu and Wilson, 2013*]) input from negative-valence MBONs that promote turning (MBON-h1, MBON-h2, MBON-i1, *Figure 4a*). KC input is also thought to be cholinergic (*Barnstedt et al., 2016*).

To test whether MBON-m1 is activated by an innately attractive odor via the LH pathway, we compared MBON-m1 calcium responses to ethyl acetate in animals with silenced KCs (by expressing tetanus toxin light chain with GMR14H06-LexA> LexAop-TNTe) (*Sweeney et al., 1995*) and controls with functional KCs (*Figure 4b*). Since PN42a/42b do not significantly synapse onto any neurons other than the KCs and the LHNs shown in *Figure 3e*, any activation or inhibition of MBON-m1 would have to be mediated by these LHNs when KCs are silenced. We verified that the MB pathway is indeed silenced by this method by observing no odor memory after odor-sugar training (see Materials and methods, *Figure 4—figure supplement 1a-c*). We imaged MBON-m1 activity in intact animals immobilized in a microfluidic device (for improved imaging we used first rather the) (*Si et al., 2019*). We did this in untrained individuals who had never been exposed to specific associative olfactory training. We found that, on average, across a population of untrained animals, MBON-m1 was activated by ethyl-acetate, both in the presence, and absence of a functional MB pathway (*Figure 4c*, *Figure 4—figure supplement 2a-b*, *Supplementary file 2*). This indicates that LHNs provide functional excitatory input to MBON-m1.

Interestingly, odor-evoked responses were highly variable across individuals (ranging from inhibition to excitation), even though, on average across individuals the response was excitatory (*Figure 4c*). ANOVA revealed that inter-individual variability was significantly higher than the intra-individual variability (i.e. high and significant Fisher's F in *Supplementary file 2*). Compared to those with silenced KCs, inter-animal variability in animals with a functional MB contributed to a higher fraction of the overall variance in odor response (*Figure 4c*, and compare the $r^2$ coefficients in *Supplementary file 2*). This suggests that the MB could be a significant source of inter-individual variability in odor-evoked responses in MBON-m1.

## MBON-m1 receives functionally excitatory and inhibitory connections from the MB

Next, we asked whether the observed anatomical pathway between the MB and MBON-m1 is functional. Unfortunately, the LH neurons being a more diverse cell population than the KCs, no driver

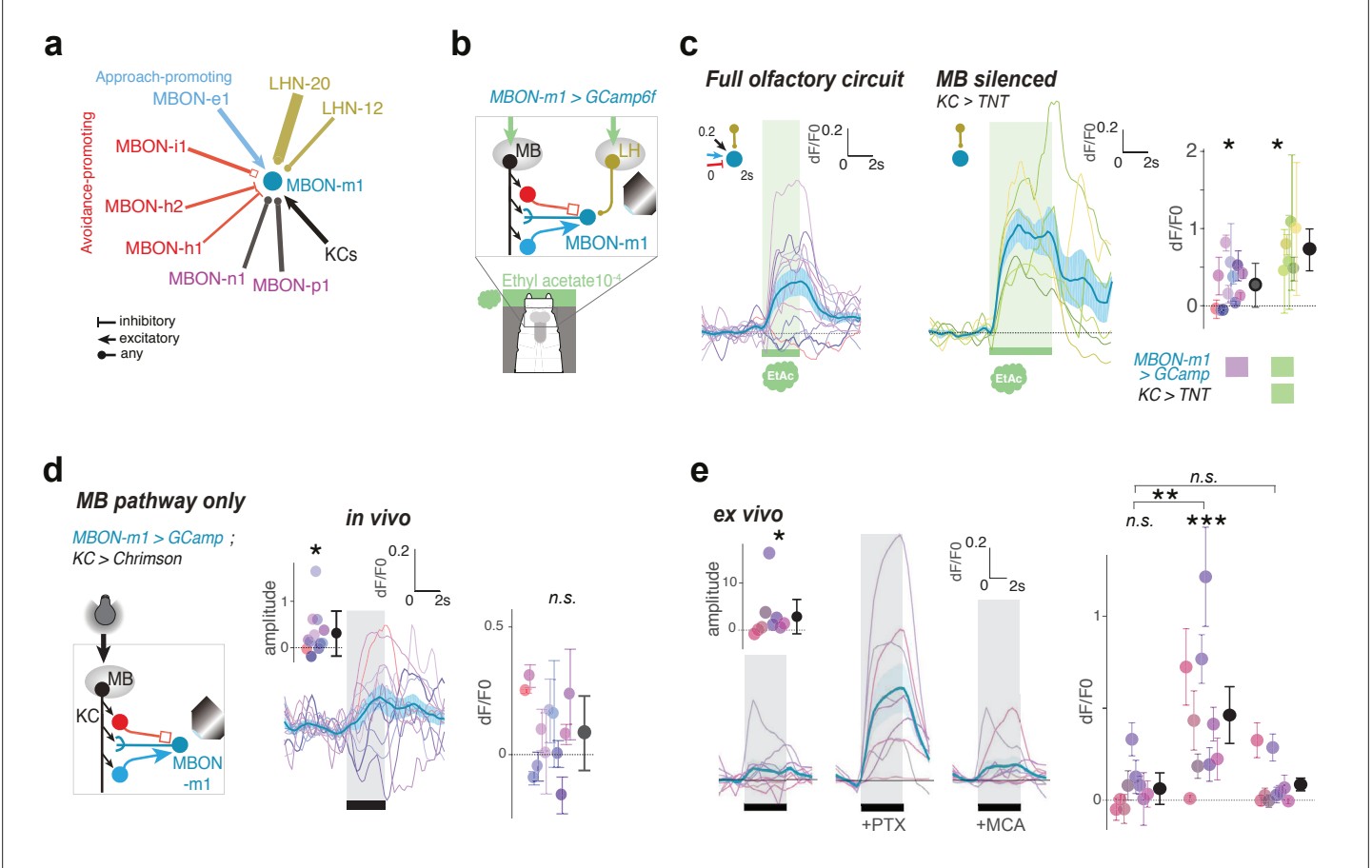

**Figure 4.** MBON-m1 receives functional inputs from LH and MB. (**a**) MBON and LHN inputs onto MBON-m1 revealed by EM reconstruction. (**b**) Calcium activity of MBON-m1 in response to an innately attractive odor (EA) (**c**) and/or to optogenetic activation of CsChrimson-expressing KCs (**d**) was imaged in vivo in first instar larvae immobilised in a microfluidic chip as in *Si et al., 2019*. (**c–e**) Plots show fluorescence normalized to baseline (prior to odor presentation, δ F/F0). Scores are calculated for the first 3 s of odor presentation. *: p < 0.05, **: p < 0.01, ***: p < 0.001, Wilcoxon test. See *Figure 4— figure supplement 2* for individual animal responses. (**c**) *Left*, In untrained larvae MBON-m1 is excited by the innately attractive odor (N = 12). *Middle*, Silencing the MB pathway (by expressing TNTe in KCs) does not impair the excitatory response of MBON-m1 to the innately attractive odor (N = 6). This confirms a functionally excitatory connection from LHNs to MBON-m1. *Right*, quantification shows significant MBON-m1 responses to odor in the presence (*left*) and absence (*middle*) of a functional MB pathway. The comparison suggests that in the naive animals the excitatory odor drive could come mainly from the LH pathway. (**d**) Across the population of untrained larvae, the average change in δ F/F0 in response to optogenetic activation of KCs is not significantly different from 0. However, the amplitude of δ F/F0 (|max δ F/F0 - min δ F/F0|) was significantly increased after, compared to before, KC activation, suggesting that MBON-m1 receives functionally excitatory inputs from the MB in some individuals, and inhibitory in others (N = 12). (**e**) Activation of KCs in brain explants together with pharmacological blockers confirm mixed inhibitory and excitatory inputs from the MB pathway (N = 9). In saline, MBON-m1 response to KC activation does not differ from zero, with excitatory or inhibitory responses depending on individuals (inset shows the amplitudes of the responses significantly differ from zero). However, when bathed with PTX, thereby blocking chloride gating GABA and glutamate receptors, MBON-m1 is robustly excited by KC activation. Bath with MCA mostly abolishes the response to KC activation.

The online version of this article includes the following figure supplement(s) for figure 4:

**Source data 1.** Individual GCaMP fluorescence values in response to odour presentation by MBON-m1 of animals with intact olfactory pathway.

**Source data 2.** Individual GCaMP fluorescence values in response to odour presentation by MBON-m1 of animals with silenced MB.

**Source data 3.** Individual GCaMP fluorescence values in response to Kenyon cells optogenetic activation by MBON-m1.

**Figure supplement 1.** Silencing KC impairs olfactory memory performances but maintains olfactory perception.

**Figure supplement 2.** Calcium responses of MBON-m1 for each individual.

lines exist that target the LH neurons as a population. Instead, we optogenetically directly activated the KCs, thus bypassing the LH (using GMR14H06-LexA line to drive CsChrimson) and imaged calcium transients in MBON-m1 (*Figure 4d*). We did this in untrained individuals. Optogenetic activation of KCs evoked diverse responses in MBON-m1, from inhibitory to excitatory or no response (*Figure 4d*,

*Figure 4—figure supplement 2c*). We note that a natural odor is expected to activate only a small fraction (*ca.* 5% *Honegger et al., 2011*) of KCs. In our experiments with direct optogenetic activation of all KCs, MBON-m1 could receive a stronger excitatory input from KCs than in response to a natural odor. Despite this, KC activation did not activate MBON-m1 in many individuals, and on average, across the population of untrained animals, there was no significant excitatory response (i.e. the mean $\partial F/F_0$ is not significantly different from zero). Albeit variable, a response to KCs was generally present in MBON-m1: overall, the amplitude of change of $\partial F/F_0$ (i.e. |maximum $\partial F/F_0$ - minimum $\partial F/F_0$ |) was significantly higher after, compared to before, KCs activation. This finding is consistent with a response which is sometimes excitatory and sometimes inhibitory across individuals (*Figure 4d*, right).

We found that the inter-individual variability of MBON-m1 responses to KC activation was significantly higher than the intra-individual variability both for in vivo and ex vivo assays (see Fisher's F in *Supplementary file 2*). The variability in the response to KC activation between individuals could be due to a number of factors, including slightly different levels of CsChrimson expression, different states or different experiences.

Our observation that MBON-m1 has excitatory response to KC activation in some individuals, inhibitory response in some, or no response in others suggests it receives both excitatory and inhibitory inputs from the MB pathway. This is consistent with the connectome which shows MBON-m1 receives direct synaptic input both from inhibitory (GABA or glutamatergic), avoidance-promoting MBONs and from excitatory (cholinergic), approach-promoting MBONs, as well as from the KCs (*Figure 4a*). To further confirm that both the excitatory and inhibitory inputs from the MB pathway onto MBON-m1 are functional, we activated KCs and recorded MBON-m1 responses in explants, either in the absence or in the presence of blockers of inhibitory or excitatory neurotransmitter receptors (*Figure 4e*). Specifically, we bathed the same sample sequentially with saline, then with the chloride-gating GABA/glutamate receptor blocker, picrotoxin (PTX, 100;;M [*Liu and Wilson, 2013*]) and then with the nicotinic acetylcholine receptor blocker, mecamylamine (MCA, 100;;M *Liu and Wilson, 2013*). A strong excitatory response to KC activation appeared when PTX was added to the solution and disappeared when MCA was added to the solution. These results demonstrate that MBON-m1 receives both functional inhibitory and excitatory inputs from the MB pathway, in accordance with the direct synaptic input it receives from GABAergic, glutamatergic, and cholinergic MBONs, as well as from KCs. These functional connections could also be mediated via MB2ONs that are downstream of these MBONs. We speculate that learning could modulate the balance of excitation and inhibition onto MBON-m1 by modifying the strength of direct KC input onto MBON-m1 as well as KC input onto excitatory or inhibitory MBONs that synapse onto MBON-m1.

## Aversive earning depresses MBON-M1 response to conditioned odors

We wanted to confirm that learning can modify MBON-m1 conditioned-odor response. MBON-m1 receives direct input from DANs that are activated by aversive stimuli and whose activation paired with odor induces aversive memory (*Eschbach et al., 2020*). Based on studies in adult *Drosophila* (*Séjourné et al., 2011*; *Hige et al., 2015a*), we predicted that aversive learning would reduce the conditioned-odor drive to the positive-valence MBON-m1.

To test this, we asked whether pairing an innately attractive odor with punishment (optogenetic activation of Basin interneurons) (*Ohyama et al., 2015*) depresses the response of MBON-m1 to that odor (*Figure 5a*, *Figure 5—figure supplement 1*). We performed these experiments in first instar larvae immobilized in a microfluidic device (*Figure 5a*; *Si et al., 2019*). We found that the response to conditioned odor was variable across individuals (*Figure 4—figure supplement 2d*), but on average, we found significantly decreased responses of MBON-m1 to the paired (CS+) and not the unpaired odor (CS-, *Figure 5a*, *Supplementary file 2*). In some individuals we also observed an increased response to the CS- after training (*Figure 5a*, *Figure 4—figure supplement 2d*). In previous studies temporally pairing Basin activation with an odor in freely behaving animals induced an aversive memory where the mean learning score across multiple batches of animals was –0.3 (i.e. on average, the number of larvae approaching the odour was decreased by 30%) (*Eschbach et al., 2020*). Also, an appetitive associative memory has been shown to be formed to the unpaired odor in larvae undergoing aversive odor training (*Schleyer et al., 2018*). Thus, our results after training immobilised animals in the microscope, showing a dramatic decrease in MBON-m1 response to CS+

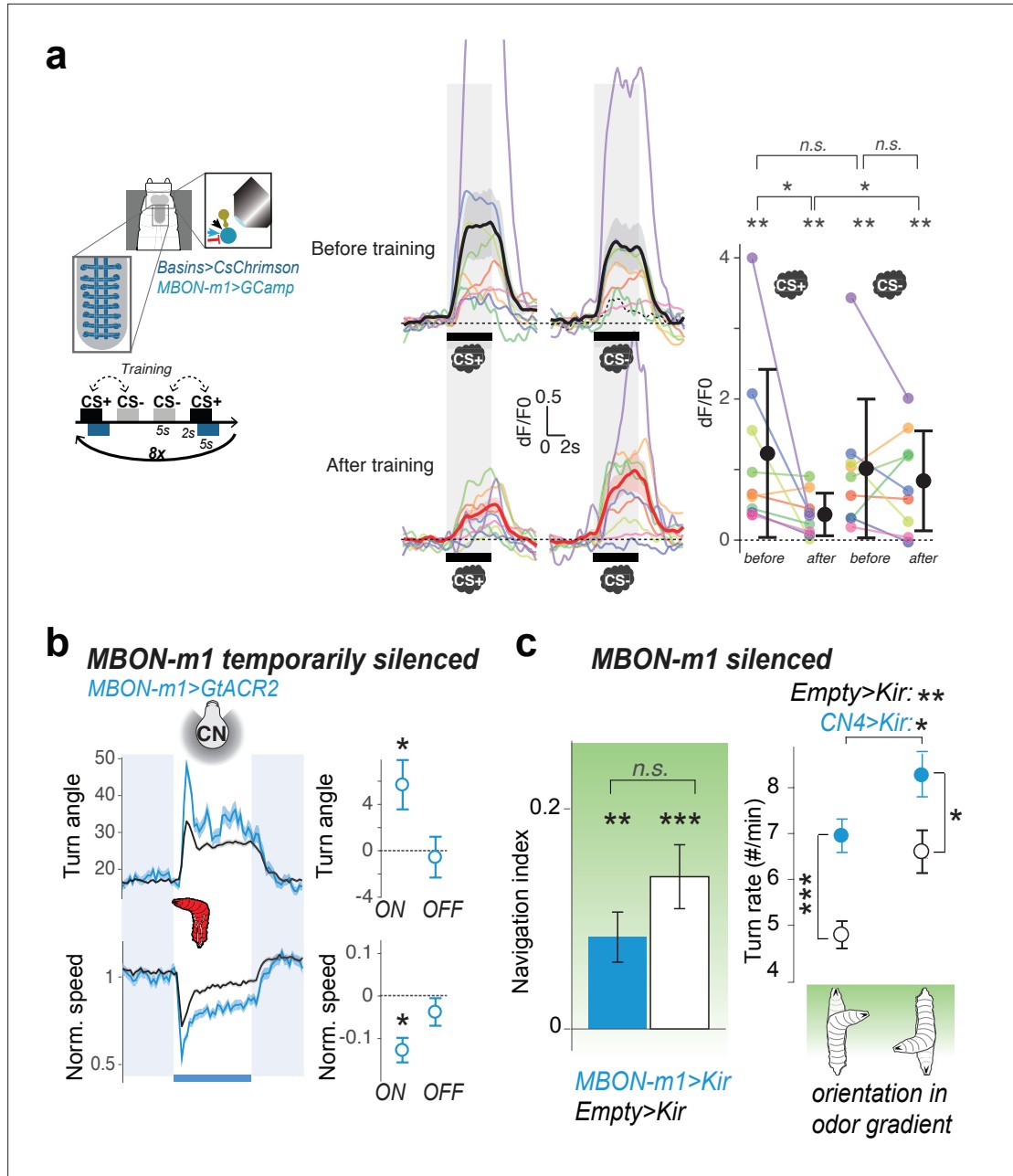

**Figure 5.** MBON-m1 activity regulates approach and is modified by experience. (**a**) Calcium activity of MBON-m1 was imaged in vivo (as in *Figure 4b*), before and aversive after training. Fictive punishment (optogenetic activation of Basins) was delivered 2 s after the exposure to the paired odor (CS+). Another odor (CS-) was presented without punishment. The response to CS+, but not to CS-, significantly decreased after training (N = 9). Curves show average fluorescence normalized to baseline (i.e. before odor presentation)+/− s.e.m. Plots show mean fluorescence scores during odor delivery per animal, and their average+/− s.e.m. in black. The odors presented were different for different animals and were one of the following 4 combinations of CS+/CS- odors: n-amyl acetate AM+/EA- (N = 4), EA+/AM- (N = 3), EA+/ME- (N = 1), ME+/EA- (N = 1) (where AM: n-amyl acetate $10^{-3}$, EA: ethyl acetate $10^{-4}$, ME: methanol $10^{-1}$). *: p < 0.05, **: p < 0.01 in a paired Wilcoxon test. (**b**) Acute optogenetic silencing of MBON-m1 (with GtACR2, $N_{exp}$ = 4, $N_{larvae}$ = 95; control empty line: $N_{exp}$ = 20, $N_{larvae}$ = 700) increases turning, in contrast to its activation which reduces turning (*Figure 2f*). Plot and quantification as in *Figure 2b–g*. *: p < 0.05 in Welch's Z test. (**c**) Constitutively silencing MBON-m1 (with Kir2.1, $N_{exp}$ = 9; control empty line: $N_{exp}$ = 10) also induced an increase in turn frequency during odor navigation (in a gradient of ethyl acetate), consistent with acute inactivation results (*left panel*), *: p < 0.05, **: p < 0.001, ***: p < 0.0001 in Welch's Z test. We also observed a reduction in the approach of the attractive odor source, likely due to a reduced ability to modulate turn rate as a function of orientation in odor gradient, but the effect did not reach significance (*right panel*), n.s.: p > 0.05.

The online version of this article includes the following figure supplement(s) for figure 5:

**Source data 1.** GCaMP responses to presentation of unpaired odour by MBON-m1 in animals after pairing odour and aversive stimulation.

*Figure 5 continued on next page*

*Figure 5 continued*

**Source data 2.** GCaMP responses to presentation of paired odour by MBON-m1 in animals after pairing odour and aversive stimulation.

**Source data 3.** Value angles showing turn responses of animals during optogenetic silencing of MBON-m1.

**Figure supplement 1.** Learning performances in first-instar larvae.

in 3 out of 9 animals, and an increased response to CS- in another three animals (*Figure 5a*), are by and large consistent with learning scores in freely behaving animals.

Aversive learning could reduce the conditioned-odor drive to MBON-m1 in two ways: (i) through direct depression of the KC-to-MBON-m1 connections due to the pairing of KC activation with DAN-g1 and -d1 activation (activated by punishment *Eschbach et al., 2020*; (ii) aversive learning in other compartments could disinhibit MBONs that inhibit MBON-m1 (*i.e.* reduced activation of positive-valence MBONs-g1/g2 would disinhibit negative-valence MBON-i1, *Figure 2i*)).

## MBON-M1 bi-directionally controls turning

Finally, we wanted to investigate in more detail the effect of MBON-m1 activity on behaviour. We have already found that optogenetic activation of MBON-m1 represses turning, consistent with the idea that it signals positive valence (*Figure 2f*). In contrast, we observed the opposite response at the offset of optogenetic activation (*Figure 2f*), suggesting that a decrease in MBON-m1 activity relative to the baseline may promote turning. To confirm this, we acutely, optogenetically hyperpolarized MBON-m1 with the anion channel-rhodopsin GtACR2 (*Govorunova et al., 2015*; *Mohammad et al., 2017*) and found this increased turning (*Figure 5b*). Thus, increasing and decreasing MBON-m1 activity relative to the baseline was sufficient to decrease and increase turning, respectively.

We also asked whether constitutively hyperpolarized MBON-m1 would have an effect on turn rate and odor navigation. We expressed Kir2.1 in groups of untrained larvae and recorded their behavior in a gradient of an innately attractive odor, ethyl acetate (*Figure 5c*). Indeed, we found that silencing MBON-m1 resulted in increased turn-rate compared to controls, consistent with the acute inactivation results (*Figure 5c*). While this can reduce the overall ability of larvae to modulate turn-rate depending on orientation in odor gradient, the animals still managed to approach the odor source (*Figure 5c*). Thus, constitutively inactivating just this one neuron type did not significantly impair odor navigation, possibly due to the existence of 10 other similar, partially redundant CN types.

## CN-33 is excited by innately attractive odors via the LH pathway

MBON-m1 is just one of a population of 10 CNs that integrate input from positive- and negative-valence MBONs, and whose activation is predicted to potentially encode positive valence and suppress turning (*Figure 3f*). We wanted to functionally investigate at least one other member of this class and confirm the predictions that (1) it is activated by the innately attractive odor via the LH pathway, (2) it receives both functional excitatory and inhibitory input from the MB, and (3) it represses turning. We had a Split-GAL4 line (*Pfeiffer et al., 2010*), SS02108 (*Eschbach et al., 2020*), that drives gene expression in a CN that we called CN-33/FAN-7 (*Figure 6a–f*). This neuron was previously shown to provide feedback to DANs, although its LH inputs were unknown (*Eschbach et al., 2020*). Like MBON-m1, it receives cholinergic input from positive-valence MBON-e1, whose activation represses turning and glutamatergic (likely inhibitory) input from negative-valence MBONs whose activation promotes turning (MBON-i1 and MBON-e2, *Figure 6a*). It also receives input from the same two LHNs downstream of PN42a/PN42b as MBON-m1 (*Figure 6a*).

To functionally test the contributions of the LH and MB to CN-33 activity, we performed the same kinds of imaging experiments as we did for MBON-m1 (*Figure 6b–d*). We compared CN-33 calcium responses to ethyl acetate in animals with silenced KCs and controls with functional KCs (*Figure 6b*, *Figure 6—figure supplement 1a-b*). We imaged CN-33 activity in intact, untrained animals immobilized in a microfluidics device. We found that, on average, in untrained animals, CN-33 was activated by ethyl-acetate, both in the presence, and absence, of a functional MB pathway (*Figure 6b*). This indicates that the LHN inputs onto CN-33 are functional, and that the innately attractive odor activates CN-33 via the LHN pathway, in untrained animals, as was the case for MBON-m1.

As was the case for MBON-m1, ANOVA revealed that inter-individual variability in CN-33 odor response was significantly higher than the intra-individual variability (i.e. high and significant Fisher's F

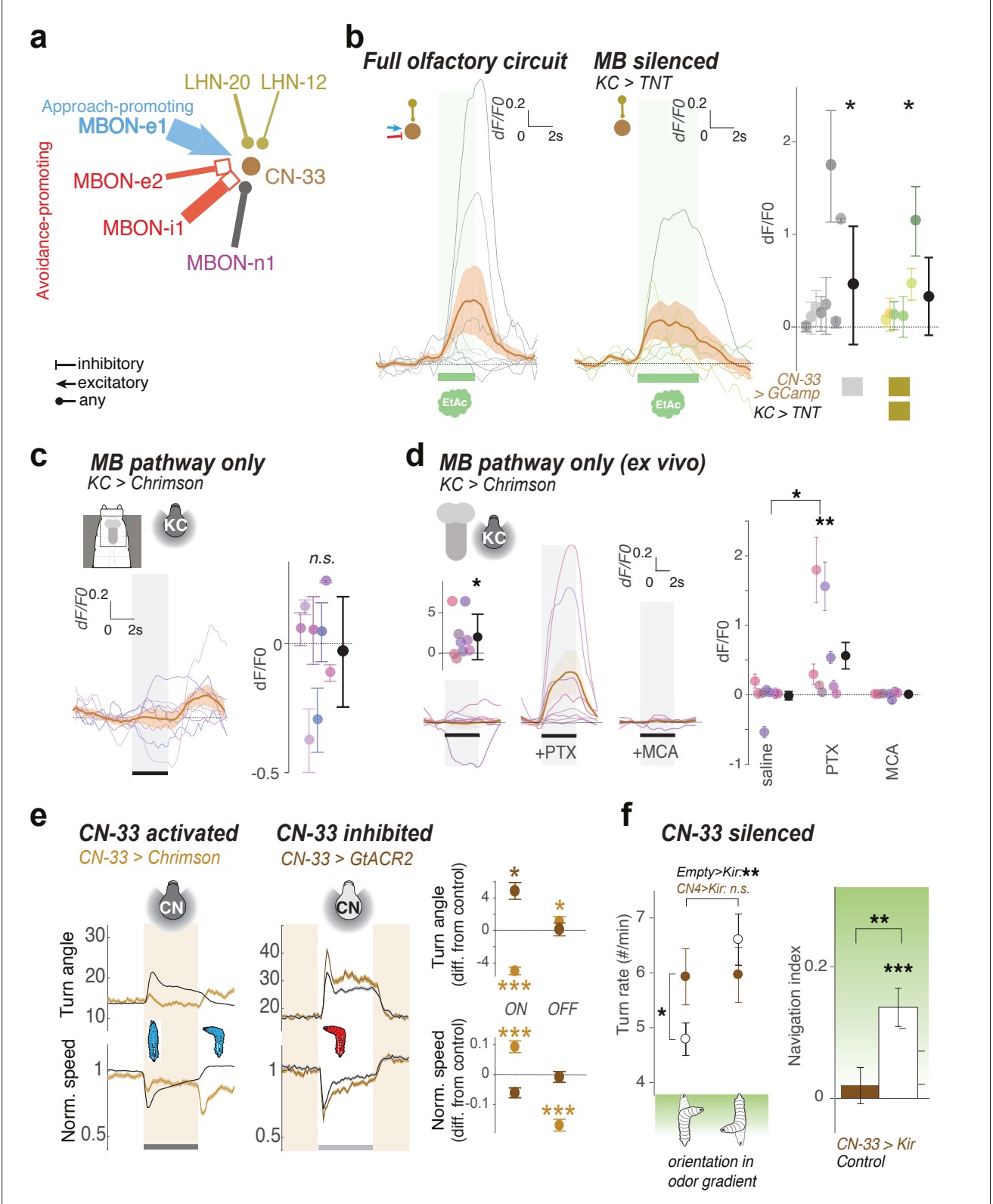

**Figure 6.** CN-33 integrates functional LH and MB inputs and regulates approach. (**a**) MBON and LHN inputs onto CN-33 revealed by EM reconstruction. (**b–d**) Calcium responses of CN-33 in untrained larvae to an innately attractive odor (EA) (**b**), or to optogenetic activation of KCs (**c–d**). Plots show fluorescence normalized to baseline (before odor presentation, δ F/F0) and scores are calculated for the first 3 s of odor presentation. Individual animal traces are shown in *Figure 6—figure supplement 1*. *: p < 0.05, **: p < 0.01 in Wilcoxon test. (**b**) Calcium responses of CN-33 to an innately attractive

*Figure 6 continued on next page*

*Figure 6 continued*

odor (EA) was imaged in vivo as in **Figure 4b–c**, either with the MB pathway intact (*left*) or blocked (by expressing TNTe in KCs) (*middle*). *Left*, On average, in untrained larvae, CN-33 is excited by the innately attractive odor (N = 8). *Middle*, Silencing the MB pathway does not impair the excitatory response of CN-33 to the innately attractive odor (N = 6). This confirms a functionally excitatory connection from LHNs to CN-33. *Right*, quantification shows no significant difference between CN-33 responses to odor in the presence (*left*) and absence (*middle*) of a functional MB pathway, suggesting in naive animals the excitatory odor drive could come mainly from the LH. (**c–d**) CN-33 responses to optogenetic activation of KCs in vivo (**c**) and in brain explant (**d**) confirm functional excitatory and inhibitory connections from MBONs to CN-33. c. Across the population of untrained larvae the average change in δ F/F0 in response to optogenetic activation of KCs does not differ from zero. We observed a slight excitation at light offset, possibly due to inhibitory rebound. Also, the amplitude of δ F/F0 (|max δ F/F0 - min δ F/F0|) was significantly increased after, compared to before, KC activation, suggesting that, similar to MBON-m1, CN-33 receives functionally excitatory inputs from the MB in some individuals, and inhibitory in others (N = 8). (**d**) Activation of KCs in brain explants together with pharmacological blockers confirm mixed inhibitory and excitatory inputs from the MB pathway (N = 8) onto CN-3. While responses to KC do not differ from zero in saline or MCA bath, blocking inhibitory signal integration with PTX unravels excitatory inputs derived from MBONs. (**e**) Optogenetic activation of CN-33 (as in **Figure 2b–g**, $N_{exp}$ = 11, $N_{larvae}$ = 400; control empty line: $N_{exp}$ = 20, $N_{larvae}$ = 700) leads to decreased and increased turning at onset and offset of activation, respectively (*left*). Acute optogenetic inactivation of CN-33 (as in **Figure 5b**) increases turning (*middle*). Plots and quantification (*right*) as in **Figure 2b–g**. See **Figure 6—figure supplement 2** for additional controls. (**f**) Constitutively silencing CN-33 (with Kir2.1, $N_{exp}$ = 8, $N_{larvae}$ = 146; control empty line: $N_{exp}$ = 10) abolished the ability of larvae to modulate turn rate as a function of orientation in odor gradient (*left panel*) and significantly impairs approach of an innately attractive odor in untrained animals (*right panel*). *: $p < 0.05$, **: $p < 0.001$, ***: $p < 0.0001$ in Welch's Z test. See **Figure 6—figure supplement 2** for additional controls. *: $p < 0.05$, ***: $p < 0.001$ in Welch's Z test.

The online version of this article includes the following source data and figure supplement(s) for figure 6:

**Source data 1.** Individual GCaMP fluorescence values in response to odour presentation by CN-33 of animals with intact olfactory pathway.

**Source data 2.** Individual GCaMP fluorescence values in response to odour presentation by CN-33 of animals with silenced mushroom body.

**Source data 3.** Individual GCaMP fluorescence values in response to Kenyon cells optogenetic activation by CN-33.

**Source data 4.** Value angles showing turn responses of animals during optogenetic silencing of CN-33.

**Figure supplement 1.** Calcium responses of CN-33 for each individual.

**Figure supplement 2.** Control crosses in activation or silencing experiments confirm that CN-33 is involved in approach behavior.

**Figure supplement 2—source data 1.** Value angles showing turn responses of control lines for CN-33 activation.

in **Supplementary file 2**). In addition, compared to those with silenced KCs, inter-animal variability in animals with a functional MB contributed to a higher fraction of the overall variance in odor response (i.e. compare the $r^2$ coefficients in **Supplementary file 2**). The MB could therefore be a significant source of inter-individual variability in odor-evoked responses of CN-33.

## CN-33 receives functionally excitatory and inhibitory connections from the MB

To confirm a functional connection from the MB onto CN-33 we optogenetically activated all KCs in untrained individuals and imaged calcium transients in CN-33. We did this either in intact living animals immobilized in the microfluidics device (**Figure 6c**, **Figure 6—figure supplement 1c**), or in extracted central nervous systems (**Figure 6d**). Both in vivo or ex vivo, the optogenetic activation of KCs evoked inhibitory responses in some individuals, excitatory responses in some, or no response in others (**Figure 6c and d**). As was the case for MBON-m1, on average, across the population of untrained animals, KC activation did not significantly activate CN-33 (**Figure 6c**), but the absolute amplitude of the responses was increased after KC activation as compared to before.

CN-33 receives direct synaptic input both from inhibitory avoidance-promoting and from excitatory approach-promoting MBONs, which could counterbalance each other. Consistent with this anatomical connectivity, adding the GABA/vGluT receptor blocker, PTX, revealed strong excitatory responses of CN-33 to KC activation. This experiment confirms that CN-33 receives functional inhibitory and excitatory input from KCs (**Figure 6d**). Since CN-33 does not receive any direct KC input but does receive direct MBON input, these functional connections must be indirect and mediated by the MBONs (and possibly also by MB2ONs that receive input from these MBONs and synapse onto CN-33).

As for MBON-m1, we found high interindividual variability of response to KCs activation compared to intraindividual variability (*i.e.* significant Fisher's F, **Supplementary file 2**), which suggests that the balance of excitatory *vs.* inhibitory drive from MB pathways onto CN-33 is individual-specific.

Together, the structural and functional connectivity patterns support the idea that CN-33, like MBON-m1, integrates functionally excitatory and inhibitory MB inputs to compute learned odor

valence and integrates this resulting valence with the innate positive valence signalled by the LH pathway (*Figure 6b*).

## CN-33 bi-directionally controls turning and contributes to odor approach

Our results thus far suggest that CN-33 activation signals positive valence. If this is the case, we would expect that activating CN-33 will repress turning (allowing odor approach). To test this, we asked whether optogenetically increasing the activity of CN-33 would repress turning (using SS02108> UAS-CsChrimson). SS02108 drives expression in two neurons with similar morphology in each hemisphere, CN-33 and MB2ON-86, and weakly in segmentally repeated local interneurons in the nerve cord downstream of aversive mechanosensory neurons. As a control, we removed SS02108-driven nerve cord expression using the Split-GAL4 repressor Killer Zipper (*Dolan et al., 2017*) under the control of teashirt-LexA (J.-M. Knapp and J. Simpson, unpublished data) promoter (*Figure 6—figure supplement 2a-b*). As a second control, we also activated MB2ON-86 alone using SS04330 (see *Eschbach et al., 2020*, *Figure 6—figure supplement 2C-d*). We found that optogenetic activation of CN-33 decreased turning and increased crawling (*Figure 6e*, *Figure 6—figure supplement 2a-d*). Additionally, we observed increased turning at the activation offset (*Figure 6e*). To confirm this another way, we acutely optogenetically hyperpolarized CN-33 with GtACR2 (*Govorunova et al., 2015*; *Mohammad et al., 2017*) and found this increased turning relative to controls (*Figure 6e*). Thus, as was the case for MBON-m1, increasing and decreasing CN-33 bi-directionally modulates turning. To confirm that this effect was not specific to light-induced turning, we additionally performed reverse correlation analysis between turning behavior and red-light intensity derivative in larvae expressing CsChrimson in CN-33 and exposed to variable intensity of red light (that controlled CsChrimson activity) under a constant blue light (that masked the red light, *Figure 2—figure supplement 3*). This allowed us to confirm that decreased activity if CN-33 correlated with turning behavior. Furthermore, the greater the decrease in CN-33 activity during a turn, the higher the probability of rejecting that turn direction and performing another turn (*Figure 2—figure supplement 3*).

Finally, we also asked whether constitutively inactivating CN-33 directly affected odor navigation. We constitutively hyperpolarized CN-33 by expressing Kir2.1 in groups of untrained larvae using *SS02108* and recorded their behavior in a gradient of ethyl acetate (*Figure 6f*). These larvae completely lost the ability to modulate turning as a function of their orientation in the odor gradient and showed a significant reduction in their ability to approach the attractive odor source (*Figure 6f*, see *Figure 6—figure supplement 2e* for control line *SS04330*). These results suggest that CN-33 contributes to odor approach by modulating turning.

## Discussion

Selecting whether to approach or avoid specific cues in the environment is essential for survival across the animal kingdom. Many cues have both innate valences acquired through evolution and learned valences acquired through experience that can guide action selection (*Rangel et al., 2008*). Innate and learned valences are thought to be computed in distinct brain areas (*Li and Liberles, 2015*; *Choi et al., 2011*; *Sosulski et al., 2011*; *Marin et al., 2002*), but the circuit mechanisms by which they are integrated and by which learned valences can override innate ones are poorly understood. Using the tractable *Drosophila* larva as a model system, we describe with synaptic resolution the patterns of convergence between the output neurons of a learning center (the MB) and an innately attractive pathway in the lateral horn (the LH). We identified 62 neurons per brain hemisphere that represent direct points of convergence between the MB and the LH, that fall into a number of different subtypes based on their patterns of MB and LH inputs and potentially encode a number of distinct features (*Figure 3e*). One subtype of 10 convergence neurons (CNs) receives excitatory input from positive-valence MB and LH pathways and inhibitory input from negative-valence MB pathways. We confirmed functional connectivity from LH and MB pathways and behavioral roles of two of these neurons. These neurons encode integrated odor value (coding for positive value with an increase in their activity) and regulate turning. They are activated by an attractive odor, and when activated they repress turning. Conversely, when inactivated, they increase turning. Based on this, we speculate that learning could potentially skew the balance of excitation and inhibition onto these neurons and thereby modulate

turning. For one of these neurons, we indeed verified that aversive learning skews inputs towards inhibition. Together, our study provides insights into the circuit mechanisms by which learned valences could interact with innate ones to modify behavior.

## Patterns of onvergence between brain areas encoding innate and learned valences

The brain areas that compute innate and learned valences of stimuli interact with each other (*Sosulski et al., 2011*; *Root et al., 2014*; *Heisenberg, 2003*; *Schleyer et al., 2011*; *Wystrach et al., 2016*), but despite recent progress, their patterns of interaction are not fully understood (*Dolan et al., 2019*; *Dolan et al., 2018*; *Scaplen et al., 2020*; *Bates et al., 2020*; *Schlegel et al., 2021*; *Li et al., 2020*). In principle, MBONs could synapse directly onto LHNs thereby directly modifying innate valences. Alternatively, LHNs could directly synapse onto MBONs. Finally, learned and innate valences could initially be kept separate, and MBONs and LHNs could converge on downstream neurons. We have found that in *Drosophila* larva (i) some MBONs synapse directly onto some LHNs; (ii) some MBONs received direct synaptic input from LHNs and (iii) many MBONs and LHNs converge onto downstream CNs (*Figure 3e*), similar to findings in the adult (*Dolan et al., 2019*; *Dolan et al., 2018*; *Schlegel et al., 2021*). One MBON (m1) was also a CN, receiving significant direct input from other MBONs and from LHNs (*Figures 3e and 4a*). Overall, the architecture suggests some early mixing of representations of innate and learned valences, but to some extent these representations are also kept separate in the LH and MB, and then integrated by the downstream CNs. Maintaining some initial separation of representations of innate and learned valences prior to integration could offer more flexibility and independent regulation, for example by context or internal state.

## Convergence neurons could compute learned valence by comparing odor-drive to positive- and negative-valence MBONs

The prevailing model of MB function in adult *Drosophila* proposes that in naive animals the odor drive to positive- and negative-valence MBONs is equal such that their outputs cancel each other out (*Owald and Waddell, 2015*) and the LH circuits guide olfactory behavior (*Li and Liberles, 2015*; *Heimbeck et al., 2001*; *Dolan et al., 2019*; *Lerner et al., 2020*). Learning alters the overall output towards positive- or negative-valence MBONs by modifying specific KC-to-MBON connections. This model raises several important questions. First, how is the output from MBONs of opposite valence integrated to compute a learned valence? Second, how does it interact with the output of the LH? Our findings provide further support for this model and shed insight into these questions.

Our EM reconstruction combined with neurotransmitter information has revealed a class of 10 CNs that receive excitatory input from positive-valence MBONs and inhibitory input from negative-valence MBONs (*Figure 3f*). These CNs are well poised to compute the learned odor valence by comparing the odor drive to positive- and negative-valence MBONs.

For two members of this class (MBON-m1 and CN-33), we have tested their MB drive in untrained larvae, in vivo or in explants. On average, across the population of untrained individuals, KC activation did not induce a significant change in $\partial F/F_0$ in either CN (*Figures 4d and 6c*). However, we found that in some individuals the MB drive onto the CNs was excitatory, and in others inhibitory, indicating that the MB can provide both excitatory and inhibitory drive to the CNs (*Figures 4d and 6c*, *Figure 4—figure supplement 2c* and *Figure 6—figure supplement 1c*). Consistently, when inhibitory neurotransmission was blocked using PTX, MBON-m1 and CN-33 showed strong excitatory responses to KC activation (*Figures 4e and 6d*).

We observed high variability in the responses of both CN-33 and MBON-m1 both to activation of the whole MB pathway, as well as to odor presentation. This variability may in part be due to technical reasons. Indeed, the variable responses to pan-KC optogenetic activation could be due to differences in CsChrimson expression in KCs across individuals (we verified the expression of CsChrimson for each individual recording but did not quantify it). However, such a difference is unlikely to be a reason for a range of responses that spans inhibition to excitation following the stimulation of the same neuronal population. Likewise, the exposure of the sensory organ to an odor may vary from one animal to the other depending on the location of the larva's head in the chip's channel. Together with other technical aspects, this may account for some of the variance observed in in vivo. Indeed, a decomposition of the variance within the datasets by ANOVA revealed a significant effect of the

identity of the individual on the variance (see *Supplementary file 2*). Interestingly though, the inter-individual variability contributed to a higher fraction of the overall variance in odor response in larvae with functional MBs than in larvae with silenced MBs (see $R^2$ values in the *Supplementary file 2*). This analysis suggests that the MB pathway is a significant source of variability in odor-evoked responses in MBON-m1 and CN-33 and is consistent with the highly variable responses of these CNs to direct KC activation. The variability in response of CNs to MB inputs across untrained individuals could be due to different experiences prior to these experiments, as suggested also by the fact that MBON-m1 responses to trained odors are modified with training (*Figure 5a*). High variability in MBON responses to odor across individuals has previously also been observed in the adult flies and has been related to different individual experiences (*Hige et al., 2015b*). The MBs can also compute other kinds of information, such as internal state (*Krashes et al., 2009*; *Bräcker et al., 2013*; *Cohn et al., 2015*; *Tsao et al., 2018*; *Siju et al., 2020*; *Modi et al., 2020*) which may modulate an individual's disposition toward sensory stimuli. Therefore, the average response across untrained individuals might be similar to the response of a single individual in a naive neutral state (with any interindividual differences averaged out across a population), and the variability may represent the degree of freedom for the MB to tune this response to a stimulus depending on previous experience or state.

## Convergence neurons integrate learned and innate valence and regulate turning

How do the learned valences modify the innate ones? How is conflict between opposite innate and learned valences resolved during action selection? One possibility could involve the integration of conflicting signals into a unified representation of value, a notion similar to common currency valuation of options (*Pearson et al., 2014*; *Levy and Glimcher, 2012*), which could then be used to promote or suppress specific actions (*e.g. Paisios et al., 2017*; *Wystrach et al., 2016*). Our findings suggest that the 10 CNs that can read out the learned valence by comparing MBON inputs of opposite valence also integrate the learned valence with the innate one. Thus, the 10 CNs that receive inhibitory input from negative-valence MBONs and excitatory input from positive-valence MBONs, also receive inputs from a positive-valence LHN pathway (*Figure 3f*). These neurons are therefore well poised to compute an integrated odor value and code for a positive value with an increase in their activity. For two members of this class (MBON-m1 and CN-33) we were able to confirm these predictions. We have shown they are activated by an innately attractive odor via the LH pathway in untrained animals (*Figures 4c and 6b*). Furthermore, when the innate attractiveness of an odor was reduced through aversive training, the activity of MBON-m1 was also reduced (*Figure 5a*). Finally, we have shown that activating MBON-m1 and CN-33 represses turning (*Figures 2f and 6e*), further supporting the idea that their activation encodes positive value. Interestingly, we also found that a decrease in their activity promotes turning (*Figures 2f, 5b–c , and 6e–f*), raising the possibility that they could bi-directionally encode value, coding for negative value with a decrease in their activity.

In principle, a single CN of this type could potentially be sufficient to compute integrated odor value and regulate turning, but we found a population of 10 CNs with similar patterns of input from positive- and negative-valence pathways that likely operate partially redundantly with each other. Each CN also had unique aspects of connectivity, raising the possibility that they may also encode partially complementary features and that the integrated value could be distributed across the CN population.

## Learning can modulate turning response to odors by modulating CN activity

Based on our findings we propose the following model that could explain the way in which learning could modulate innate responses to odors through a population of integrative CNs (*Figure 7*): (i) In naive animals, the CNs are activated by innately attractive odors, mainly via the LH pathway (as shown in *Figures 4c and 6b* for untrained animals); (iii) when activated these neurons repress turning (*Figures 2f and 6e*), which enables crawling towards an attractive odor source; (iv) in naive animals the net MB drive onto these CNs is close to 0 (as shown in *Figures 4d–e , and 6c–d* across a population of untrained animals); (v) aversive learning can skew the net MB output onto CNs towards inhibition, so that aversively-conditioned odor fails to activate these neurons (as shown for MBON-m1, *Figure 5a*); (vii) if CNs are not activated, turning rate remains high when crawling towards the odor

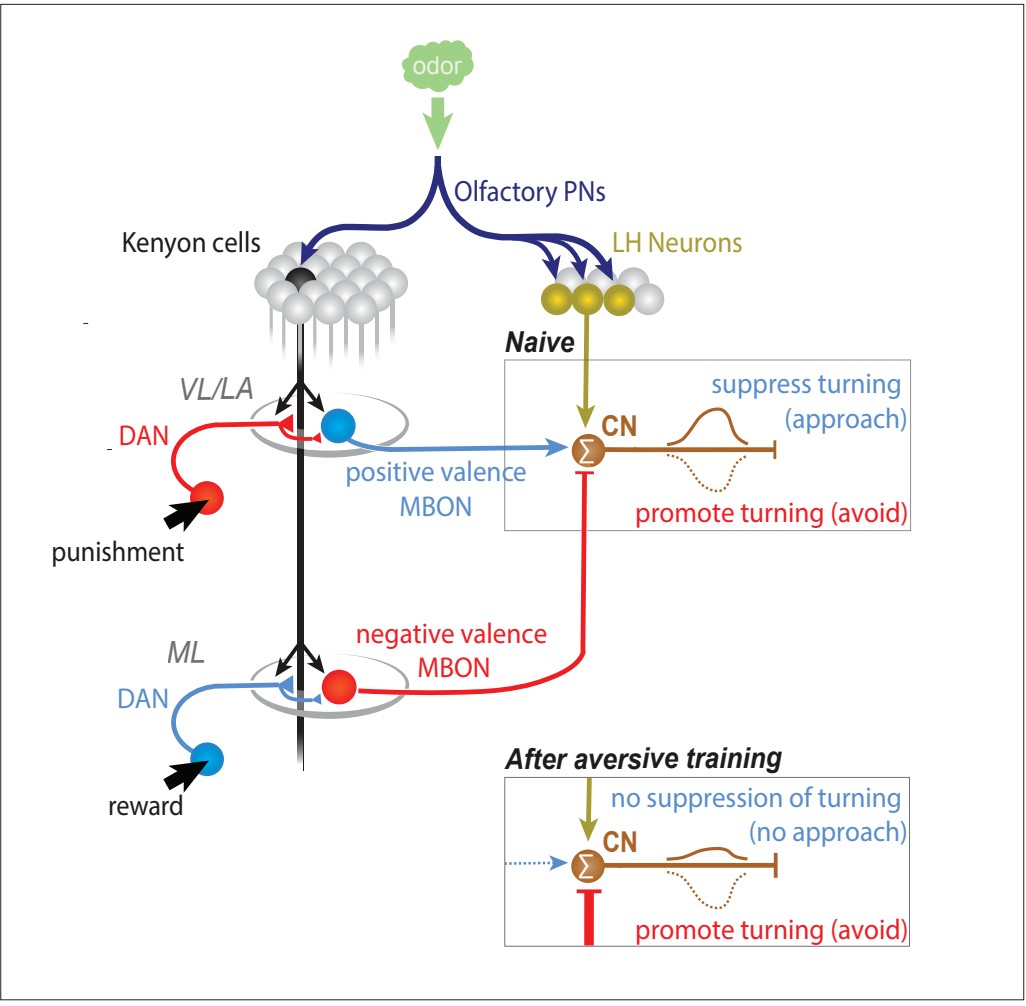

**Figure 7.** CN-33 integrates LH and MB-derived inputs. CNs whose activation suppresses turning are activated by innately attractive odors in naive animals via the LH pathway (when the net MB output onto them is 0). Learning could bidirectionally skew the net MB output onto the CNs towards excitation (appetitive learning), or inhibition (aversive learning). After aversive learning the conditioned odor would activate these CNs less, failing to suppress turning, or sometimes even inhibit them, thereby inducing turning and inducing avoidance.

(see *Figures 5b–c , and 6e–f*, for turning rate when CNs are silenced); (viii) without sufficient suppression of turning by an odor, the animals' ability to approach the odor source is impaired (*Figure 6f*). This proposed model could explain how aversive learning suppresses approach of innately attractive odors. To fully suppress approach, multiple CNs of this class would likely need to be silenced.

The model presented above could readily be extended to explain how appetitive learning could enhance odor approach and how strong aversive learning could switch innate odor approach to learned avoidance. Thus, appetitive learning could skew the net MB drive towards excitation thereby repressing turning even more. In contrast, strong aversive learning could skew the conditioned odor drive towards inhibitory negative-valence MBONs so much that the inhibition would become stronger than the excitatory LH drive. The CN would then be inhibited by the aversively conditioned odor. Since inhibition of CNs promotes turning, the aversively conditioned odor could promote odor avoidance by inducing turning. Consistent with this idea, we observed inhibitory responses to the innately attractive odor in MBON-m1 and in CN-33, in some untrained individuals, and strong excitatory responses in others (*Figures 4c and 6b*), proving that odor drive onto CNs can range all the way from strong excitation to inhibition. We did not observe inhibition of MBON-m1 following aversive learning in the microfluidic device, but this could be because the animals did not form a very strong aversive memory under these conditions. Testing these proposed extensions of the model will require

the future development of automated single-animal training assays and calcium imaging tracking microscopes to correlate the strength of the learned behavior with the conditioned-odor response of the CNs.

Finally, our EM reconstruction did reveal a potentially opposite population of CNs that are inhibited by MBONs of positive value (*Figure 3e–f*). It will be interesting to test in the future whether these neurons potentially encode negative value with an increase in their activity and positive value with a decrease in their activity and regulate turning in the opposite way: promoting turning when they are activated and repressing it when they are inhibited. Having two different populations of neurons that encode value in opposite ways could further increase the dynamic range of a distributed value code.

## A behavior signal feeds back onto MB modulatory neurons

We find that many of the CNs that receive input from both MBONs and LHNs also provide direct (including CN-33) or indirect (including MBON-m1) feedback to MB modulatory neurons that provide teaching signals for learning (n = 18). In a former study (*Eschbach et al., 2020*), we have shown that at least some of these feedback connections are functional and can influence memory formation. For example, CN-33/FAN-7 is capable of generating an olfactory memory when it is paired with an odor. This type of connectivity is consistent with learning theories that propose that future learning is influenced by predicted value computed based on prior learning (*Schultz, 2015*; *Cognigni et al., 2018*). A major role of the CNs discovered in this study may therefore be not only to organize current actions, but also to regulate learning.

In summary, the comprehensive synaptic-resolution architecture of the circuits downstream of the learning center output neurons presented in this study is a valuable resource for constraining future modelling and function studies of value computation, action selection, and learning.

# Materials and methods

## Larvae

Larvae were reared in the dark at 25 °C in food vials. The food was supplemented with trans-retinal (SIGMA R2500) at a final concentration of 500 µM if the genome contained the *UAS-CsChrimson* transgene. For behavior experiments, the larvae were selected at their third-instar stage; for imaging experiments, they were at first-instar. We verified that first-instar larvae were capable of learning using optogenetic manipulations (*Figure 5—figure supplement 1*).

To train larvae with optogenetic punishment, we used the *GMR72F11-GAL4* (*Ohyama et al., 2015*; BDSC 39786) line crossed to *20XUAS-CsChrimson-mVenus* (*Klapoetke et al., 2014*; BDSC 55134). To assess the response to optogenetic activation of MBONs or CNs, Split-GAL4 (*Pfeiffer et al., 2010*) were crossed to direct the expression of *20XUAS-CsChrimson-mVenus*. The empty stock *y w;attP40;attP2* (*Pfeiffer et al., 2010*) was crossed to the effector as a baseline control. The following driver lines were used:

| Neuron | SplitGAL4 | AD | DBD | Figure | Reference |
|---|---|---|---|---|---|
| MBON-a1 | *SS02006* | *93* G12 | *71E06* | 2 | unpublished |
| MBON-a2 | *SS01417* | *52E12* | *93* G12 | 2 | unpublished |
| MBON-b1+ MBON-b2 | *SS01708* | *12* G03 | *21D02* | 2 | unpublished |
| MBON-c1 | *SS01776* | *74B11* | *20* F01 | 2 | unpublished |
| MBON-d1 | *SS01705* | *11E07* | *52* H01 | 2 | unpublished |
| MBON-d2 | *SS04231* | *121* A04 | *87* G02 | 2 | unpublished |
| MBON-d3 | *SS24027* | *111B05* | *100* H11 | 2 | unpublished |
| MBON-e2 | *SS04559* | *65* A05 | *102D01* | 2 | unpublished |
| MBON-g1 or g2 | *SS02130* | *23B09* | *21D06* | 2 | unpublished |
| MBON-h1 or h2 | *SS00894* | *67B01* | *11* F03 | 2 | unpublished |

*Continued on next page*

*Continued*

| Neuron | SplitGAL4 | AD | DBD | Figure | Reference |
|---|---|---|---|---|---|
| MBON-i1 | *SS01726* | *20* C05 | *14* C08 | 2 | *Eschbach et al., 2020* |
| MBON-j1 | *SS01972* | *128* F10 | *12* C11 | 2 | *Eschbach et al., 2020* |
| MBON-k1 | *SS01962* | *VT033301* | *27* G01 | 2 | *Saumweber et al., 2018* |
| MBON-m1 | *SS02163* | *52* H01 | *40* F09 | two and 6 | *Eschbach et al., 2020* |
| CN-4+ MB2ON-86 | *SS02108* | *13D05* | *40* F09 | four and 5 | *Eschbach et al., 2020* |
| MB2ON-86 | *SS04330* | *17* H07 | *40* F09 | Ext. data | *Eschbach et al., 2020* |

For a few MBONs, GAL4 lines inserted at the *attP2* site (from the FlyLight GAL4 collection), (*Jenett et al., 2012*) were crossed to *UAS-CsChrimson; tsh-GAL80*, to antagonize effector expression in the ventral nerve cord. For some lines, tsh-LexA and LexAop-KZip+::3xHA was used to antagonize Split-GAL4 expression (*Dolan et al., 2017*).

*Tsh-LexA* was a gift from J.-M. Knapp and J. Simpson (unpublished stock). In brief, the *tsh-LexA* driver is an enhancer trap inserted into the 5' UTR of the tsh locus. It was generated via a P-element swap that replaced the *p{GawB}* insertion of *tsh-GAL4* (*Calleja et al., 1996*) with *P{UpP65L}* and the enhancer-trap LexA construct. Proper targeting and orientation of *P{UpP65L}* were confirmed by splinkerette PCR and sequencing by J.-M. Knapp (unpublished results).

The empty stock *yw;;attP2* (*Jenett et al., 2012*) was also crossed to *UAS-CsChrimson; tsh-GAL80* as a baseline control. The following driver lines were used:

| Neuron | GAL4 driver | Figure | Reference |
|---|---|---|---|
| MBON-e1 | *GMR74B11* | 2 | *Saumweber et al., 2018* |

To test whether the response to the activation of the olfactory neuron Or42b requested functional Kenyon cells, we made the following constructs: (1) effector stock: *13XLexAop2-CsChrimson-mVenus* (in *attP18*, *Klapoetke et al., 2014*; BDSC 55137); *UAS-Kir2.1::GFP* (gift from Y. Aso), crossed to (2) driver stock: *w + CS; Or42b-LexA* (in JK22C), (*Fishilevich and Vosshall, 2005*; *GMR14H06-GAL4* (in *attP2*, *Jenett et al., 2012*; BDSC 48667)). The effector stock was crossed to the pan-neuronal driver *GMR57C10-GAL4* (*Jenett et al., 2012*; BDSC 80962) to verify Kir2.1 was blocking neurons (no larvae hatched from the crosses); and to the empty line *y w;attP40;attP2* (*Pfeiffer et al., 2010*) to have a reference response to optogenetic activation of Or42b. A second reference response was obtained by crossing the driver stock to the line *13XLexAop2-CsChrimson-mVenus* (in *attP18*; *Klapoetke et al., 2014*; BDSC 55137).

To further characterize the properties of CN-4 and MBON-m1, we crossed the Split-GAL4 lines SS02108 (for CN-4) and SS02163f (for MBON-m1) to the following constructs:

- *UAS-Kir2.1* (*Baines et al., 2001*) to hyperpolarize CNs and test their contribution in navigation behavior in a gradient of ethyl acetate.
- *UAS-GtACR2* (gift from V. Jayaraman and A. Claridge-Chang); (*Govorunova et al., 2015*; *Mohammad et al., 2017*) to observe behavioral response to acute hyperpolarization of the CNs.
- *20XUAS-IVS-Syn21-opGCaMP6f p10 Chen et al., 2013*; *14H06-LexAp65* in *JK22C*; *Jenett et al., 2012*; *LexAop-TNTe* (gift from J. Simpson, unpublished line; *Sweeney et al., 1995*) to image CNs' response to odors while silencing MB pathway. This stock was also crossed to pan-neuronal line *GMR57C10-LexA* to verify TNTe was silencing neurons (no larvae hatched from the crosses). In addition, we directly verified that KC output was indeed blocked by training the *SS02108* experimental cross in odor-sugar pairing and observing no immediate memory as compared to a control wild-type group (*Figure 4—figure supplement 1*).
- *pJFRC22-10xUAS-IVS-myr::tdTomato* (in *su(Hw)attP8*; *Pfeiffer et al., 2010*; BDSC 32223); *72F11-LexAp65* in *JK22c*; *Jenett et al., 2012*; *20xUAS-GCaMP6f 15.693* (in *attP2*, *Chen et al., 2013*), *13xLexAop2-CsChrimson-tdTomato* (in *vk00005*; *Klapoetke et al., 2014*; BDSC 82183) to image CNs' response to odors before and after aversive training. To verify that larvae resulting from such crosses were capable of learning, we trained first-instar larvae that resulted from a

cross between *SS02108* and the above line, following a pairing protocol between odor and optogenetic activation. The larvae fed with retinal showed strong aversive short-term memory, whereas the larvae without retinal did not (see *Figure 5—figure supplement 1* for details).

## Learning experiments and odor navigation

Learning experiments were performed as previously described (*Eichler et al., 2017*; *Eschbach et al., 2020*; *Saumweber et al., 2018*). Briefly, two groups of 30 third-instar larvae were separated from food and underwent a training procedure involving odor and light exposures, either fully overlapping in time (paired group), or fully non-overlapping (unpaired group). The paired group was placed for 3 min on 4 % agarose plates and exposed to constant red-light illumination (629 nm, 2.5 μW/mm2) paired with the presentation of 12 μl of odor ethyl-acetate 4.10$^{-6}$ dilution in distilled water) absorbed on two filter papers located on the plate lid. These larvae were then transferred to a new plate with no odor and in the dark for 3 minutes. This paired training cycle was repeated three times in total. The unpaired group of larvae underwent odor presentation in the dark and red light without odor following the same protocol. These larvae were then immediately transferred to a 25 cm$^2$ custom-made odor-delivery arena (described in *Gershow et al., 2012*) covered with 4 % agar and illuminated with infrared (for detection) and red (for memory expression, *Gerber and Hendel, 2006*; 2.5 μW / mm2) light. A linear gradient of odor was generated by modulating the opening times of 32 3-way valves. One input to each valve was fed with odorized air (generated by bubbling air through a 4.10$^{-6}$ dilution of EtAc in distilled water and the other with humidified air (generated by bubbling air through distilled water). The outlet of each valve projected into one of 32 parallel channels pushing humidi-fied air across the chamber; a microcontroller and custom electronics were used to switch the valve opening times to produce a linear gradient (*Gershow et al., 2012*). The navigation behavior of the larvae was recorded with a camera (DALSA Falcon 4M30) and analyzed with machine vision developed in Matlab.

To test the role of CNs in innate odor navigation we placed larvae expressing Kir2.1 in neurons or control larvae in a gradient of odor. *Ca*. 30 larvae were separated from food, rinsed, and placed in an 25 × 25 cm square dish filled with 4 % agar and whose one side of the lid was covered with five glued filter papers (7 mm$^2$) each loaded with 12 μl of 10$^{-4}$ ethyl acetate. The navigation behavior of the larvae was recorded with a camera (DALSA Falcon 4M30) and analyzed with machine vision (Matlab).

For all navigation behaviors, bouts of individual trajectories were reconstructed offline, and inter-rupted when detection was compromised (e.g. if the larvae reached the border or crosses another larva's path). To quantify the overall navigational response in the linear spatial gradient, we computed a navigation index by dividing the mean velocity of all larvae in the *x*-direction by the mean crawling speed. Hence, the navigational index was +/−1 if the larvae crawled uniformly straight up/down the gradient and 0 if the movement was unbiased. For each bout and at each time point, the position of the larva relative to the odor gradient was estimated as well as its likelihood to be engaged in a turn (details in *Gershow et al., 2012*). To quantify turn-based navigation strategy, we compared the turn probability when the larvae were aligned towards the gradient (±15 degrees) to the turn probability when the larvae were aligned away from the gradient (±15 degrees). The experiments were repeated 7–10 times in each condition. The mean and s.e.m. of these parameters were computed for each experiment and further pooled for the repeats. A Welch z test was used for statistical comparisons.

## Optogenetic neural activation screen

Optogenetic activation experiments were performed as previously described (*Ohyama et al., 2015*; *Jovanic et al., 2016*). *Ca*. 30 larvae were separated from food by bathing them in a 20 % sucrose solu-tion for a maximum of 10 min. They were rinsed and placed into a square 23 cm$^2$ behavior rig covered with 4 % agar. We recorded videos of larval behavior, with a DALSA Falcon 4M30 camera for a total of 120 s. At 30 and 75 s, 15 s-long pulses of red 660 nm red light (4 μW/mm2, Philips Lumileds) were applied. For statistics, speed (normalized to the baseline before the stimulation) and turn angle are averaged for all larvae and time windows corresponding to the two stimulation. For onset response, a time window was taken during 5 s after light on for offset response, during 10 s after light off. Welch z tests were used for statistical comparisons.

Larvae were tracked in real-time using the MultiWormTracker (MWT) software (*Swierczek et al., 2011*; *Ohyama et al., 2013*; *Vogelstein et al., 2014*). We rejected objects that were tracked for less

than 5 s or moved less than one body length of the larva. For each larva MWT returns a contour, spine and center of mass as a function of time. Raw videos are never stored. From the MWT tracking data, we computed the key parameters of larval motion, using specific choreography (part of the MWT software package) variables. Turn angle was defined and computed as in *Ohyama et al., 2013*; *Vogelstein et al., 2014*. Briefly turn angle (deg) is the absolute value of the distance from the least-squares line fit of the posterior 2/3 of the animal's spine points (typically 7 points out of 11) to the point in the anterior 1/5 of the animal's spine (typically 2 points out of 11) most distant from that line.

## Light-turn experiments for reverse correlation

Detailed protocol is described in *Gepner et al., 2015*. In brief, *ca.* 30–50 larvae were transferred to a 23 cm square dish containing 2.5 % (wt/vol) agar and 0.75 % activated charcoal (to improve contrast for detection). The plate was placed in a darkened enclosure and larvae were observed under strobed 850 nm infrared illumination (ODL-300–850, Smart Vision Lights, Muskegon, MI) using a 14 fps 5 MP rolling shutter CMOS camera (Basler acA2500-14gm, Graftek Imaging, Austin, TX) in global-reset-release mode and an 18 mm c-mount lens (54–857, Edmund Optics, Barrington, NJ) equipped with an IR-pass filter (Hoya R-72, Edmund Optics). For light stimulation, a custom circuit board (Advanced Circuits, Colorado) containing 66 deep red high brightness LEDs (Philips Lumileds, LXM3-PD01, 655 nm) and 12 royal blue high brightness LEDs (LXML-PR01-0500, 447.5 nm) evenly distributed over ~25 cm × 25 cm. The LEDs were driven at constant current by a switch-mode LED driver circuit (based on LT3518, linear technology) operating at a switching frequency of 2 MHz. Illumination was provided from above the larvae; the LED circuit board was at the same height as the recording camera (~50 cm above the behavioral arena). The intensity of the red and blue LEDs was controlled separately by pulse-width-modulation; in these experiments, blue LEDs was set at a constant intensity (3.7 μW/cm2) and the red light intensity varied following a Brownian random walk at an update rate of 112 Hz, and where light levels were specified by values between 0 (off) and 255 (maximum intensity, i.e. 911 μW/cm2). CsChrimson is slightly more sensitive to 655 nm than 448 nm light, so the blue light signal perturbed olfactory receptor neuron activity by less than 0.3 % of the red-light signal's perturbation. Videos were recorded using custom software written in LABVIEW and analyzed using the MAGAT analyzer software package (*Gershow et al., 2012*). Further analysis was carried out using custom MATLAB scripts. Software is available at https://github.com/GershowLab.

Turn-triggered averages (TTA) with a bin size of 0.1 s were computed by averaging stimulus values at the corresponding times relative to the start of a turn. We smoothed the TTA by fitting it to the impulse response of a third order linear system used to describe the calcium dynamics of *Caenorhabditis elegans* olfactory neurons (*Kato et al., 2014*). We used the smoothed TTA as a convolution kernel to find the output of the linear filter stage of a Linear non Poisson model. We scaled the kernel so that the variance of the filtered signal over the entire stimulus history was 1. The turn rate $r(x_f)$ and standard error $\sigma_r(x_f)$ as a function of filtered signal value ($x_f$) were computed by

$$r\left(x_f\right) = \frac{N_{turn}\left(x_f\right)}{N_{all}\left(x_f\right)} * \frac{1}{\Delta t}.$$

$$\sigma_r\left(x_f\right) = \frac{\sqrt{N_{turn}\left(x_f\right)}}{N_{all}\left(x_f\right)} * \frac{1}{\Delta t}.$$

$N_{turn}$ is the number of turns observed with the filtered signal within the histogram bin (size = 0.25) containing $x_f$ and $N_{all}$ is the total number of data points where the filtered signal was in the histogram bin and larvae were in runs and thus capable of initiating turns, and $\Delta t$ was the sampling period (1/14 s).

By construction, the stimulus ensemble is Gaussian distributed with mean 0 and variance 1. If the turn-triggered ensemble is also Gaussian distributed, then the turn rate is given by a ratio-of-Gaussians (*Schwartz et al., 2006*).

$$r_{ROG}(x) = \bar{r}\frac{e^{-\frac{(x-\mu)^2}{2\sigma^2}}}{\sigma e^{-\frac{x^2}{2}}}; \bar{r} = \frac{N_{turn}}{T}; \mu = E\left[x_f \mid turn\right]; \sigma^2 = E\left[\left(x_f - \mu\right)^2 \mid turn\right].$$

T is the total time the larvae were in runs and therefore able to initiate turns. $r_{ROG}(x)$ is a rate function, with $\underline{r}$, $\sigma$, and $\mu$ calculated directly from the turn-triggered ensemble.

For head sweep triggered averages, analysis was conducted as for the TTA, but the reference time (0) was chosen as the beginning of either rejected or accepted head sweeps. Because the decision to accept or reject a head sweep is made after the beginning of the head sweep, we expected that the average would be nonzero at positive times corresponding to the duration of a head sweep. To simplify interpretation of the resulting averages, we considered only the first head sweep of each turn.

## Circuit mapping and electron microscopy

We reconstructed neurons and annotated synapses in a single, complete central nervous system from a 6-hr-old female *Canton S G1 x w1118* larva, acquired with serial section transmission EM at a resolution of 3.8 × 3.8 x 50 nm, that was first published along with the detailed sample preparation protocol (*Ohyama et al., 2015*). Briefly, the CNS was dissected and placed in 2 % gluteraldehyde 0.1 M sodium cacodylate buffer (pH 7.4). An equal volume of 2 % $OsO_4$ was added and the larva was fixed with a Pelco BioWave microwave oven with 350 W, 375 W and 400 W pulses for 30 s each, separated by 60 s pauses, and followed by another round of microwaving but with 1 % $OsO_4$ solution in the same buffer. Next, samples were stained en bloc with 1 % uranyl acetate in water and microwaved at 350 W for 3 × 3 30 s with 60 s pauses. Samples were dehydrated in an ethanol series, transferred to propylene oxide, and infiltrated and embedded with Epon resin. After sectioning the volume with a Leica UC6 ultramicrotome, sections were imaged semi-automatically with Leginon (*Suloway et al., 2005*) driving an FEI Spirit TEM (Hillsboro, OR), and then assembled with TrakEM2 (*Cardona et al., 2012*) using the elastic method (*Saalfeld et al., 2012*). The volume is available at https://l1em.catmaid.virtualflybrain.org/?pid=1.

To map the wiring diagram we used the web-based software CATMAID (*Saalfeld et al., 2009*), updated with a novel suite of neuron skeletonization and analysis tools (*Schneider-Mizell et al., 2016*), and applied the iterative reconstruction method (*Schneider-Mizell et al., 2016*). All annotated synapses in this wiring diagram fulfill the four following criteria of mature synapses (*Ohyama et al., 2015*; *Schneider-Mizell et al., 2016*) (1) There is a clearly visible T-bar or ribbon on at least two adjacent z-sections. (2) There are multiple vesicles immediately adjacent to the T-bar or ribbon. (3) There is a cleft between the presynaptic and the postsynaptic neurites, visible as a dark-light-dark parallel line. (4) There are postsynaptic densities, visible as dark staining at the cytoplasmic side of the postsynaptic membrane.

We validated the reconstructions as previously described (*Ohyama et al., 2015*; *Schneider-Mizell et al., 2016*), a method successfully employed in multiple studies (*Ohyama et al., 2015*; *Jovanic et al., 2016*; *Berck et al., 2016*; *Schneider-Mizell et al., 2016*; *Fushiki et al., 2016*; *Goodman et al., 1981*). Briefly, in *Drosophila*, as in other insects, the gross morphology of many neurons is stereotyped and individual neurons are uniquely identifiable based on morphology (*Goodman et al., 1981*; *Bate et al., 1981*; *Costa et al., 2016*). Furthermore, the nervous system in insects is largely bilaterally symmetric and homologous, with mirror-symmetric neurons reproducibly found on the left and the right side of the animal. We therefore validated neuron reconstructions by independently reconstructing synaptic partners of homologous neurons on the left and right side of the nervous system. With this approach, we have previously estimated the false positive rate of synaptic contact detection to be 0.0167 (1 error per 60 synaptic contacts) (*Schneider-Mizell et al., 2016*). Assuming the false positives are uncorrelated, for an n-synapse connection the probability that all n are wrong (and thus that the entire connection is a false positive) occurs at a rate of $0.0167^n$. Thus, the probability that a connection is a false positive reduces dramatically with the number of synaptic contacts contributing to that connection. Even for n = 2 synaptic contacts, the probability that a connection is not true is 0.00028 (once in every 3,586 two-synapse connections). We therefore consider 'reliable' connections those for which the connections between the left and right homologous neurons have at least three synapses each and their sum is at least 10. See (*Ohyama et al., 2015*; *Schneider-Mizell et al., 2016*) for more details. When predicting valence of CNs based on input from MBONs of known neurotransmitters and behavioral effects (approach or avoidance behaviors), we required a combined input of 5 % from the appropriate MBONs to ensure that our predictions were robust.

## Similarity matrices and clustering

Adjacency matrices of synaptic connectivity were converted to binary connectivity matrices, representing only strong connections between hemilateral neuron pairs. A strong connection is defined

as at least three synapses from the presynaptic left neuron and three synapses from the presynaptic right neuron onto the postsynaptic left and right neurons and a sum of at least 10 synapses total. Ipsilateral and contralateral connections are considered. Similarity is obtained by counting indices of value one that are observed at the same location in both the row neuron pair and the column neuron pair (matches) and counting the total number of value one indices that are only observed in the row or column alone, but not both (mismatches). The similarity score is the total number of matches, divided by the total number of matches and mismatches. Hierarchical clustering of similarity matrices was performed using R and heatmap.2 {gplots}.

## Identifying GAL4 lines that drive expression in MBONs and CNs

To identify GAL4 lines that drive expression in specific neurons, we performed single-cell FlpOut experiments (for FlpOut methodology see *Ohyama et al., 2015*; *Nern et al., 2015*) of many candidate GAL4 lines (*Li et al., 2014*). We generated high-resolution confocal image stacks of individual neuron morphology (multiple examples per cell type). Most MBONs were uniquely identifiable based on the dendritic and axonal projection patterns (which MB compartment they project to and the shape of input or output arbor outside the MB). Some MBON pairs were too similar to be distinguished: MBON-h1/h2, g1/g2, and b1/b2.

## Brain explants imaging of CN response to KC optogenetic activation

Central nervous systems were dissected in a cold buffer containing 135 mM NaCl, 5 mM KCl, 4 mM MgCl2·6H2O, 2 mM CaCl2·2H2O, 5 mM TES and 36 mM sucrose, pH 7.15 (*Marley and Baines, 2011*) and adhered to poly-L-lysine (SIGMA, P1524) coated cover glass in small Sylgard (Dow Corning) plates. Picrotoxin at 0.1 mM (*Liu and Wilson, 2013*) was added to the solution to block GABA$_A$ and glutamate-gated chloride receptors. After rincing with saline for 5 min, mecamylamine at 0.1 mM was applied to the sample. Half of the samples were first exposed to mecamylamine, then to picrotoxin.

Optogenetic activation was done by red flood illumination on the sample (625 nm, Four channel LED driver, Thorlabs, power) through the objective. Light stimulations were delivered for 3 s and for four successive times (ISI *ca.* 15 s) in each scan. After recording the response to KC stimulation of the neuron bathed in the buffer, the same stimulation protocol was done once more while the sample was bathed in buffer containing 0.1 mM picrotoxin, or 0.1 mM mecamylamine (*Liu and Wilson, 2013*). Each brain was sequentially exposed to the two kinds of blockers, with rinsing with 4 ml of buffer for 3 min in between.

Scanning was done using the same two-photon system as for in vivo imaging (see below chapter). For image analysis, image data were processed using custom code in Matlab (The Mathworks, Inc). Specifically, the code automatically corrects for misaligned images, determines the regions of interest (ROIs) from maximum intensity projection of entire time series images, and measure the mean intensity of the ROIs minus the background fluorescence. In all cases, changes in fluorescence were calculated relative to baseline fluorescence levels ($F_0$) as determined by averaging over a period of 5 s. just before the optogenetic stimulation. The fluorescence values were calculated as $(F_t - F_0)/F_0$, where $F_t$ is the fluorescent mean value of a ROI in a given frame. Analyses were performed on the average of the consecutive four stimulations; comparisons of before *vs.* after stimulation were done using a non-parametric Wilcoxon test for paired comparisons and variance analyses were done using ANOVA.

## Microfluidic device design

Odorant stimuli were delivered using a microfluidic device described in detail in *Si et al., 2019* and modified to deliver three odors instead of 13. The larva loading channel was 300 μm wide and 70 μm high, and tapered to a width of 60 μm in order to immobilize the larva. The tapered end was positioned perpendicular to a stimulus delivery channel to allowing odorant to flow past larval dorsal organ that houses 21 ORNs. The device was designed with a 'shifting-flow strategy', enabling odor changes without pressure or flow rate discontinuities (*Chronis et al., 2007*). An eight-channel device included two control channels located at the periphery, three odorant channels in the middle, and one water channel to remove odorant residue (the two remaining channels were blocked by a stopper). Each channel was of equal length to ensure equal resistance. During an experiment, a combination of three channels was always open: the water channel, one of the three odorant delivery channels, and one of the control channels. The three odorant channels could be sequentially opened to deliver any

odorant. Switching between the two control channels directed either water or an odorant to flow past the larva's ORNs.

Fluorescein dye was used to verify the spatial odorant profile in the device during stimulus delivery. The air pressure for stimulus delivery was set to 3 psi, where the switching time between water and odorant was estimated to be ~20 ms.

The microfluidic device pattern was designed using AutoCAD. The design pattern was then transferred onto a silicon wafer using photolithography. The wafer was used to fabricate microfluidic devices using polydimethylsiloxane (PDMS) and following the standard soft lithography approach (*Anderson et al., 2000*). The resulting PDMS molds were cut and bonded to glass cover slips. Each microfluidic device was used for a few number of experiments and water- and air-cleaned between each of them in order to prevent contamination.

## Odorant delivery setup

Odorants were obtained from Sigma-Aldrich, diluted in deionized (DI) water (Millipore) and stored for no more than 3 days. We used n-amyl acetate (diluted in water for a $10^{-3}$ final concentration, AM), 3-octanol ($10^{-4}$, OCT), ethyl acetate ($10^{-4}$, EA), and methanol ($10^{-1}$, ME). Each odorant concentration was stored in a separate glass bottle and delivered through its own syringe and tubing set. Panels of odorants were delivered using an eight-channel pinch valve perfusion system (AutoMate Scientific Inc). Each syringe and tubing set contained a 30 mL luer lock glass syringe (VWR) connected to Tygon FEP-lined tubing (Cole-Parmer), which in turn was connected to silicone tubing (AutoMate Scientific Inc). The silicone tubing was placed through the pinch valve region of the perfusion system and could allow for the passage or blockage of fluid flow to the microfluidics device. The silicone tubing was then connected to PTFE tubing (Cole-Parmer), which was then inserted into the microfluidic device. We used a DAQ board (National Instruments) to control the eight-channel pinch valve perfusion system using custom-written MATLAB code. This custom code allowed us to implement the on/off sequence of the valves and to synchronize valve control with the onset of recording in the imaging software (ScanImage).

During the entire recording, the larva experienced continuous fluid flow. The stimuli sequences consisted of five seconds of odorant pulses followed by a washout period using water.

## In vivo calcium imaging

A first instar larva was loaded into a microfluidic device using a 1 mL syringe filled with 0.1 % triton-water solution. Using the syringe, a larva was pushed towards the end of the channel, where the 60 µm wide opening mechanically trapped further larval movement. Each larva was positioned such that its dorsal organ (nose) was exposed to the stimulus delivery channel. Larvae were imaged at 35 fps using a multiphoton microscope equipped with a fast resonant galvo scan module (customized Bergamo Multiphoton, Thorlabs) controlled by ScanImage 2016 (http://www.scanimage.org). The light source was a femtosecond pulsed laser tuned to 925 nm (Mai Tai, Spectraphysics). The objective was a 25 X water immersion objective (NA 1.1 and 2 mm WD, Nikon). The CNs neurites (dendrites and their contralateral axon terminals) were spanned in at least one brain hemisphere by a volume scan (six slices with a step size of 2 µm).

For pairing of an odor with optogenetic activation of aversive neurons, a 660 nm laser (Obis 660, Coherent) photostimulation (*ca.* 480 µW/mm$^2$) was directed towards the terminals of the aversive neurons using a galvo-galvo module (Thorlabs) controlled by ScanImage software. The scans were usually not saved during the pairing period, as the imaging laser power were set to minimum power to avoid photobleaching associated with long-run recording. The CS+ was delivered for 5 s, followed by a 5 s laser stimulation which overlapped with CS+ for 3 s. The CS- was presented alone for 5 s. Each bout consisted in two CS+ and two CS- presentation; each interspaced by 20±3 s of water flow, for a total of *ca.* 120 s-long bout. The odors presented were different for different animals and were one of the following four combinations of CS: AM+/EA-, EA+/AM-, ME+/EA-, EA+/ME-. Eight consecutive pairing bouts were performed. The position of the larva in the channel was assessed between each bout and rectified by adding triton-water-filled with the syringe if necessary. At the end of the eight bouts, the settings of the microscope were readjusted to allow optimal recording and the responses of the CN to delivery of the odors were reassessed the same way as before pairing. A single larva underwent between one and two of the pre-pairing and post-pairing scanning bouts.

The same system was used for co-stimulation of CNs with $10^{-4}$ ethyl acetate and optogenetic activation of Kenyon cells. Here, two consecutive 3 s-long odors and two consecutive 3 s-long photo-stimulations were conveyed to the larva in a shuffled order, followed by two consecutive 3 s-long joined delivery of odor and photo-stimulation. Each stimulation was interspaced with 20±2 s of water flow for a total of *ca.* 100 s-long scanning bout. A single larva underwent between one and three of these bouts.

### Odor response analyses

The GCaMP6f fluorescence (averaged intensity of z-projection) was calculated for a region of interest (CN's neurite), subtracted to background intensity, and normalized to the tdtTomato signal emitted at CN's membranes of the same region $F_t = (F_{GCaMP\_t})/ \text{median}(F_{GCaMP}) - F_{dTom\_t} / \text{median}(F_{GCaMP})$. For each larva, one to two regions of interest (corresponding to the left and right hemispheres) were selected (by thresholding the projected maximum intensity image) for each larva and their fluorescence was averaged. Two to four repetitive stimulations were averaged as well. Movement artifacts were corrected by aligning frames using the strongest signal (tdtTomato- or GCaMP6f-derived) labeling the CN neurites and a combination of cross-correlation on Matlab (normxcorr2_general, 2010, Dirk Padfield) and manual correction.

Changes in fluorescence were quantified as $\partial F/F_0 = (F_t - F_0)/F_0$, where $F_0$ was the average fluorescence intensity sampled from the frames of the 5 s preceding a stimulation. Quantifications of normalized mean and peak were the normalized value of, respectively, mean and maximum intensity for the frames during the stimulation (ON response) or for the 8 s following the stimulation (OFF response). For the absolute value of the response to KC stimulation, the difference in the absolute value of average $\partial F/F_0$ during and before KC stimulation was computed: $|\partial F/F_0$ during stimulation| - |baseline $\partial F/F_0$ prior to stimulation|.

Statistical comparisons were done using a non-parametric Wilcoxon test for paired comparisons.

### Data exclusion

When movements artefacts were too important and rendered alignment impossible for a substantial part of the recording (*ca.* 10%), the data for this larva was discarded. Data for trained larvae was excluded if no calcium response was observed to any odors before and/or after the training session, as it likely indicated that the larva was dead or that the external sensory organs were not exposed to the odor flow.

### Acknowledgements

We thank Fly Light at HHMI Janelia Research Campus (JRC) for generating confocal images of the GAL4 lines, V Jayaraman for sharing unpublished versions of CsChrimson and GCaMP6f, K Hibbard and JRC FLY Core for generating some of the fly stocks, Fly EM at JRC for generating the EM volume. We thank Feng Li (9%), Avinash Khandelwal (8%), Jennifer Lovick (5%), Ivan Larderet (5%), Timo Saumweber (5%), Volker Hartenstein (4%), Nadia Riebli (3%), Larisa Maier (2%), Alex Bates (2%), Laura Herren (1%), and many others (3%) for reconstructing arbor cable and synapses. We thank Rebecca Arruda for her contribution to behavioral data. Z Zavala-Ruiz and the JRC Visiting Scientist Program and HHMI Janelia Research Campus for funding. MZ and CE were also supported by the ERC consolidator grant LeaRNN 819650, MZ and MW were also supported by the Wellcome Trust Investigator Awards 205050/B/16/Z and 205050/C/16/Z. AC also thanks the Wellcome Trust (award 205038/Z/16/Z and 205038/A/16/Z) for funding.

# Additional information

## Funding

| Funder | Grant reference number | Author |
|---|---|---|
| Howard Hughes Medical Institute | | Claire Eschbach<br>Akira Fushiki<br>Michael Winding<br>Bruno Afonso<br>Katharina Eichler<br>Ingrid V Andrade<br>Benjamin T Cocanougher<br>Javier Valdes-Aleman<br>James W Truman<br>Albert Cardona<br>Marta Zlatic |
| European Research Council | LeaRNN 819650 | Claire Eschbach<br>Marta Zlatic |
| Wellcome Trust | 205050/B/16/Z | Marta Zlatic<br>Michael Winding |
| Wellcome Trust | 205038/Z/16/Z | Albert Cardona |
| Wellcome Trust | 205050/C/16/Z | Marta Zlatic<br>Michael Winding |
| Wellcome Trust | 205038/A/16/Z | Albert Cardona |

The funders had no role in study design, data collection and interpretation, or the decision to submit the work for publication.

## Author contributions

Claire Eschbach, Conceptualization, Data curation, Formal analysis, Investigation, Methodology, Project administration, Software, Validation, Visualization, Writing – original draft, Writing – review and editing; Akira Fushiki, Conceptualization, Data curation, Formal analysis, Investigation, Methodology, Resources, Software, Validation, Visualization, Writing – original draft; Michael Winding, Conceptualization, Data curation, Formal analysis, Investigation, Methodology, Resources, Software, Validation, Visualization, Writing – review and editing; Bruno Afonso, Conceptualization, Data curation, Methodology, Software; Ingrid V Andrade, Javier Valdes-Aleman, Data curation, Investigation; Benjamin T Cocanougher, Conceptualization, Data curation, Formal analysis, Investigation, Visualization, Writing – original draft; Katharina Eichler, Data curation, Formal analysis, Investigation, Methodology; Ruben Gepner, Data curation, Formal analysis, Investigation, Methodology, Software, Visualization; Guangwei Si, Conceptualization, Data curation, Investigation, Methodology; Richard D Fetter, Methodology, Resources; Marc Gershow, Conceptualization, Data curation, Formal analysis, Methodology, Resources, Software, Supervision, Visualization, Writing – review and editing; Gregory SXE Jefferis, Resources, Supervision; Aravinthan DT Samuel, Conceptualization, Methodology, Supervision, Writing – review and editing; James W Truman, Data curation, Investigation, Methodology, Resources; Albert Cardona, Conceptualization, Data curation, Funding acquisition, Methodology, Project administration, Resources, Software, Supervision, Writing – original draft, Writing – review and editing; Marta Zlatic, Conceptualization, Data curation, Funding acquisition, Methodology, Project administration, Supervision, Writing – original draft, Writing – review and editing

## Author ORCIDs

Claire Eschbach (iD) http://orcid.org/0000-0002-8092-3440
Akira Fushiki (iD) http://orcid.org/0000-0002-7987-6405
Michael Winding (iD) http://orcid.org/0000-0003-1965-3266
Benjamin T Cocanougher (iD) http://orcid.org/0000-0003-0648-554X
Katharina Eichler (iD) http://orcid.org/0000-0002-7833-8621
Richard D Fetter (iD) http://orcid.org/0000-0002-1558-100X
Marc Gershow (iD) http://orcid.org/0000-0001-7528-6101
Gregory SXE Jefferis (iD) http://orcid.org/0000-0002-0587-9355
Aravinthan DT Samuel (iD) http://orcid.org/0000-0002-1672-8720

James W Truman (iD) http://orcid.org/0000-0002-9209-5435
Albert Cardona (iD) http://orcid.org/0000-0003-4941-6536
Marta Zlatic (iD) http://orcid.org/0000-0002-3149-2250

**Decision letter and Author response**
Decision letter https://doi.org/10.7554/eLife.62567.sa1
Author response https://doi.org/10.7554/eLife.62567.sa2

## Additional files

**Supplementary files**
• Supplementary file 1. Adjacency matrix of all reconstructed neurons. Adjacency matrix between all neurons (all type of synaptic connection) reconstructed and considered in this and in previous studies (*Eichler et al., 2017*; *Eschbach et al., 2020*): Olfactory Projection Neurons downstream of ORN42a and 42b (PNs), Lateral Horn Neurons downstream of these PNs (LHNs), Mushroom Body Output Neurons (MBONs), MB 2nd-Order Neurons downstream of MBONs (MB2ONs), Convergence Neurons (CNs). modulatory neurons (dopaminergic, octopaminergic, or unknown: respectively DANs, OANs, MBINs), Feedback 2nd-order Neurons (FB2Ns), Feed-Forward Neurons (FFNs), Kenyon Cells (KCs).

• Supplementary file 2. Results of statistical analyses. Details of statistical analyses of imaging data: probability resulting from Wilcoxon rank test and ANOVA.

• Transparent reporting form

**Data availability**
All data generated or analysed during this study are included in the manuscript and supporting files. Source data files are provided for Figures 2 to 6.

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
