## [Editor Report]

This work of Eschbach, Fushiki and colleagues uses optogenetic approaches in the *Drosophila* larval brain to assign behavioral valence to many output neurons of the mushroom body, a brain region canonically associated with assigning learned valence to sensory stimuli. The authors present a comprehensive electron microscopic-level neuroanatomical analysis of the postsynaptic partners of mushroom body putout neurons, identifying substantial convergence from these onto neurons of the lateral horn, a region associated with innate sensory-evoked behavioral responses. Through neurophysiological and behavioral analyses of two classes of these "convergence neurons," they provide evidence for their role in representing sensory stimuli, evoking behavioral responses, and adapting after learning. Together this work provides an impressive anatomical framework, and initial functional insights, supporting the existence of extensive communication between neural circuits controlling learned and innate behavior.

---

## [Decision Letter]

**Decision letter after peer review:**

Thank you for submitting your article "Circuits for integrating learnt and innate valences in the insect brain" for consideration by *eLife*. Your article has been reviewed by 2 peer reviewers, and the evaluation has been overseen by K VijayRaghavan as the Senior Editor and Reviewing Editor. The reviewers have opted to remain anonymous.

The reviewers have discussed the reviews with one another and the Reviewing Editor has drafted this decision to help you prepare a revised submission.

Summary:

The work of Eschbach, Fushiki and colleagues is composed of three parts: first, they use optogenetic approaches in the *Drosophila* larval brain to assign behavioral valence to many output neurons of the mushroom body (MBONs), a brain region canonically associated with assigning learned valence to sensory stimuli. Second, they present a comprehensive EM-level neuroanatomical analysis of the postsynaptic partners of MBONs, identifying substantial convergence from these (as well as some MBONs) onto neurons of the lateral horn, a region associated with innate sensory-evoked behavioral responses. Third, through neurophysiological and behavioral analyses of two classes of these "convergence neurons" (CNs; one of which is also an MBON), they provide evidence for their role in representing sensory stimuli, evoking behavioral responses, and adapting after learning. Together this work provides an impressive anatomical framework, and initial functional insights, supporting the existence of extensive communication between neural circuits controlling learned and innate behavior. This work extends concepts initially established in earlier studies in adult *Drosophila*, notably valence assignment to MBONs (Aso 2014 PMID: 25535794), and MB-to-LN communication (Dolan 2018 PMID: 30244885), but has a wealth of original observations, datasets and reagents to facilitate future experimental exploration and modeling of the integration of learned and innate valence in the "simple" larval brain.

As the authors' can see from the reviews, major concerns have been expressed. In our consultations, we felt that these were important and needed to be addressed. The authors may choose to address these comprehensively and submit a revised manuscript for examination. If, on the other hand, they feel that the revisions asked will take too much time and effort to address, they may want to submit the manuscript elsewhere, In any case, please do let us know how you would like to proceed.

Essential Revisions:

*Reviewer #1:*

1. I have concerns over the strength of evidence from the calcium imaging analyses, as the response properties can be extremely variable in both magnitude and temporal dynamics between animals. I appreciate that the authors are quite transparent in displaying all their imaging traces in the supplement. However, I don't think that they are entirely objective in interpreting them. For example, analysis of MBON-m1 responses to pan-Kenyon Cell optogenetic activation can be strong, weak or non-existent, which the authors suggest is due to differences in individual animal's experience but could alternatively be due to various technical issues. (These are presumably not trivial experiments, the sample sizes for these experiments are general limited). But averaging all of the responses and then stating that "the absolute value of ∂F/F0 was significantly increased after, compared to before, KC activation … indicating that there is a response, which is sometimes excitatory and sometimes inhibitory" seems mostly guided in interpretation by the anatomical data rather than the physiological evidence obtained. In the corresponding legend (Figure 4d) it gets even more definitive "… due to the fact that MBON-m1 receives functionally excitatory inputs from the MB in some individuals, and inhibitory in others". Their interpretations are plausible but preliminary.

Similarly, a key result in Figure 5a leads the authors to claim that MBON-m1 is modified by experience. In the main figure, they (unusually) do not plot the individual traces, which prevents appreciation of the variability evident in the corresponding supplementary figure (ED Figure 9): here, out of nine animals imaged, only three show dramatically diminished odor responses post-conditioning (in several cases there is little odor response even pretraining; and the non-conditioned odor can, in some cases, also show diminished responses). None of the nuance is really captured in what the authors state: "As expected, we found significantly decreased responses of MBON-m1 to the paired (CS+) and not the unpaired odor". I have similar concerns with the physiological analysis of the other investigated neuron population (CN-33) (Figure 6c-d). For this populations, they do not examine the impact of learning (even though in the abstract they may a rather general statement about convergence neurons): "Aversive learning skews their inputs towards inhibition to modulate turning."

In sum, for such complex experiments, and complex network interactions, a much larger dataset would be needed to solidify the authors' claims. Without further experimental data, the interpretation should be significantly toned down.

2. Presentation: while the overall quality of data presentation is high, because the manuscript is both composed of several parts and enormously detailed in the neuroanatomy, the central narrative is very hard to follow. The reader has to cope with numerous acronyms and abbreviations (e.g., "some MBONs (MBON-m1) that receive direct LHN input also receive reliable input from other MBONs (Figure 3e), and are therefore also CNs (LHN-CNs and MBON-CNs), but we continue to call them LHNs and MBONs for brevity") as well as frequent laboriously constructed phrases (e.g., "a neuron whose activation signals negative valence (avoidance) is expected to promote turning in response to an increase in its activity). I realise that such linguistic turns of phrases are frequently unavoidable but in the context of such a juggernaut of data the authors could probably refine their text to make it as accessible as possible. There is a lot of repetition, including "interpretations" in the results (notably, trying to conclude the meaning of calcium imaging data in the context of the neuroanatomy – see point 1), which would be better in the Discussion, as well as unnecessary restatement of results in several of the figure legends.

*Reviewer #2:*

This manuscript uses a combination of EM-based connectomics, behavior, and calcium imaging to explore how learned and innate valence signals are integrated to drive action selection in *Drosophila*. Building on previous work where the authors characterized the valence of memory encoding for mushroom body compartments, they then characterize the valence encoding of the output neurons for each of these compartments, and reconstruct all downstream targets. They find a complex network of downstream neurons which receive input from mushroom body output neurons encoding opposite valances. Using behavior, optogenetics, and calcium imaging the authors explore how two such convergence neurons might integrate both learned and innate valence, as well as signals of opposite valence.

This work exploring the integration of valence signals in the *Drosophila* olfactory system represents an interesting parallel with the author's other recent work exploring how memory encoding in the mushroom body. Most of the conclusions are well supported. However, there are a number of issues that should be addressed prior to publication.

1. Conclusions of the paper

a. The primary hypothesis of this paper seems to be that in addition to depression of the KC-MBON connection due to aversive learning, shown in (Eschbach 2020), there exist other mechanisms for the comparing of learned valences, including integration at both the MBON and CN level. One proposed mechanism is that MBONs encoding opposite valences inhibit one another. The connectome only somewhat supports this mechanism. Of the tested MBONs, only 2 show the strict lateral inhibition described. Three have no interactions, two inhibit MBONs with both the same and opposite valence, one excites MBONs of both the same and opposite valence, and one inhibits an MBON of the same valence. Based on these connection motifs, can it really be said that lateral inhibition is a general principle for mushroom body organization?

b. The latter half of the paper focuses on two CNs, one an MBON itself, and another a feedback neuron (described in Eschbach 2020). The author's primary argument is that these neurons integrate both learned and innate valences, as well as are integrating inputs of MBONs of opposite valence. The data presented is consistent with the idea that both MBON-m1 and CN-33 integrate information from both the LH (innate) and the mushroom body (learned), however, it is not entirely clear that there is sufficient functional evidence to support the idea that either of these neurons is integrating MBON input of opposite valence. The connectome does suggest that this is the case, however, silencing of the mushroom body had no effect on the response of either MBON-m1 or CN-33 to odors. Pairing the odor with an aversive stimulus clearly demonstrates that the neurons are capable of integrating innate and learned stimuli, but it is unclear if the inputs from positive MBONs are being depressed as shown in the schematic in Figure 7.

2. Behavioral experiments

a. The optogenetic stimuli used for conditioning experiments in Figure 2 produce a large (startle?) response, on top of which all other behavior is measured. What evidence is there that the behavioral conditioning experiments in Figure 2 are not producing a potentiation or reduction of this light-induced startle response? This is particularly worrisome in the majority of cases where the dynamics of the conditioned behavior follow a similar time-course to the startle response. Overall, it is not convincing that this analysis is cleanly separating MBONs that would drive avoidance and approach independent of the light artifact.

b. Each MBON in Figure 2 is classified as driving either approach or avoidance, but the dynamics of their angular steering and speed vary widely. This should at least be discussed in the text, and the authors should justify their decision for categorization in light of this variability.

3. Calcium Imaging experiments

The calcium imaging data throughout the paper is quite noisy and responses are variable across individual animals. The authors sometimes attribute this variability to biological variability in circuit connectivity (eg, Figure 6). Where this is done, it is worth including an explicit comparison of within fly vs across variability, to test the degree to which responses are stereotyped within an animal.

---

## [Author Response]

Essential Revisions:Reviewer #1:1. I have concerns over the strength of evidence from the calcium imaging analyses, as the response properties can be extremely variable in both magnitude and temporal dynamics between animals. I appreciate that the authors are quite transparent in displaying all their imaging traces in the supplement. However, I don't think that they are entirely objective in interpreting them. For example, analysis of MBON-m1 responses to pan-Kenyon Cell optogenetic activation can be strong, weak or non-existent, which the authors suggest is due to differences in individual animal's experience but could alternatively be due to various technical issues. (These are presumably not trivial experiments, the sample sizes for these experiments are general limited). But averaging all of the responses and then stating that "the absolute value of ∂F/F0 was significantly increased after, compared to before, KC activation … indicating that there is a response, which is sometimes excitatory and sometimes inhibitory" seems mostly guided in interpretation by the anatomical data rather than the physiological evidence obtained. In the corresponding legend (Figure 4d) it gets even more definitive "… due to the fact that MBON-m1 receives functionally excitatory inputs from the MB in some individuals, and inhibitory in others". Their interpretations are plausible but preliminary.

We agree that there are alternative explanations to the variability in the response of MBON-m1 and CN-33 to KC activation, other than differences in individual experiences, including differences in state (rather than prior experience) or technical issues. To address this point further we have performed additional experiments, additional analysis of variability and modified the text to state clearly additional sources of variability.

To provide further evidence that the MBON-m1 and CN-33 do receive both indirect functional excitatory and inhibitory inputs from KCs consistent with the prediction from the connectome, we activated KCs and recorded MBON-m1 and CN-33 responses in the absence and presence of a blocker for the inhibitory chloride-gating GABA/glutamate (vGluT) receptors (picrotoxin, PTX) or acetylcholine (AChR) receptors (mecamylamine, MCA). Specifically, we expressed CsChrimson in KCs and GCamp6f in either MBONm1 or CN-33, and bathed the same sample sequentially with saline, PTX, or MCA, while activating KCs optogenetically.

In both neurons, a strong excitatory response to KC appeared when PTX was added to the solution and disappeared when MCA was added to the solution. Further the response when the neurons were bathed in saline was on average not significantly different from zero, but the overall amplitude of fluorescence change was higher during, compared to before stimulation, which suggest again a general mixed response of these neurons to KC stimulations.

These results, we think, provide conclusive evidence that MBON-m1 and CN-33 receive both functional inhibitory and excitatory inputs from the MB pathway, in accordance with the anatomical data that shows they receive synaptic input from GABAergic, glutamatergic, and cholinergic MBONs.

We have added a new Figure 4e and 6d with the new results in the presence of receptor blockers and a paragraph in the Results sections:

“Our observation that MBON-m1 has excitatory response to KC activation in some individuals, inhibitory response in some, or no response in others suggests it receives both excitatory and inhibitory inputs from the MB pathway. This is consistent with the connectome that shows MBON-m1 receives direct synaptic input both from inhibitory (GABA or Glutamatergic) avoidance-promoting MBONs and from excitatory (cholinergic) approach-promoting MBONs, as well as from the KCs (Figure 4a). To further confirm that both the excitatory and inhibitory inputs from the MB pathway onto MBON-m1 are functional, we activated KCs and recorded MBON-m1 responses in explants either in the absence or in the presence of blockers of inhibitory or excitatory neurotransmitter receptors (Figure 4e). Specifically, we bathed the same sample sequentially with saline, then with the chloride-gating GABA/glutamate receptor blocker, picrotoxin (PTX) and then with the acetylcholine (AChR) receptor blocker, mecamylamine (MCA). A strong excitatory response to KC appeared when PTX was added to the solution and disappeared when MCA was added to the solution. These results demonstrate that MBON-m1 receives both functional inhibitory and excitatory inputs from the MB pathway, in accordance with the direct synaptic input it receives from GABAergic, glutamatergic, and cholinergic MBONs, as well as from KCs.”

“To confirm a functional connection from the MB onto CN-33 we optogenetically activated all KCs in untrained individuals and imaged calcium transients in CN-33. […] Since CN-33 does not receive any direct KC input but does receive direct MBON input, these functional connections must be indirect and mediated by the MBONs (and possibly also by MB2ONs that receive input from these MBONs and synapse onto CN-33).”

To address the possibility that the variability could be due to technical issues, such as a variability in Kenyon Cell activation, we measure the intensity of the light stimulation in each experiment and we verify the expression of the TdTomato tagged CsChrimson for each recording. Nevertheless, we have also added statements in the text that the variability in the response to KC activation could be due to a number of factors: (1) slightly different levels of CsChrimson expression (2) different states and (3) different experiences.

We have also compared inter-individual variability with intra-individual variability and found that the interindividual variability is greater than intra-animal variability (Supplementary File 2). Thus, the responses of a neuron from a single animal are more consistent with each other, suggesting that at least within a single animal the conditions do not vary as much. We now display the intra-individual variability in the plots for imaging data in Figure 4c-e and 6b-d.

Interestingly, we also found that the inter-animal variability contributed to a higher fraction of the overall variance in odor response in animals with a functional MB compared to those with silenced KCs (Supplementary File 2). This suggests that the MB could be a significant source of inter-individual variability in odor-evoked responses in MBON-m1 and CN-33.

Here are the excerpts of the text that we added or modified:

Results

“Interestingly, odor-evoked responses were highly variable across individuals (ranging from inhibition to excitation), even though, on average across individuals the response was excitatory (Figure 4c). ANOVA revealed that inter-individual variability was significantly higher than the intra-individual variability (*i.e.* high and significant Fisher’s F in Supplementary File 2). Compared to those with silenced KCs, inter-animal variability in animals with a functional MB contributed to a higher fraction of the overall variance in odor response (Figure 4c, and compare the r^2^ coefficients in Supplementary File 2). This suggests that the MB could be a significant source of inter-individual variability in odor-evoked responses in MBON-m1…”

“KCs activation did not activate MBON-m1 in many individuals, and on average, across the population of untrained animals, there was no significant response (*i.e.* the mean ∂F/F0 is not significantly different from zero). […] The variability in the response to KC activation between individuals could be due to a number of factors, including slightly different levels of CsChrimson expression, different states or different experiences.”

“As was the case for MBON-m1, ANOVA revealed that inter-individual variability in CN-33 odor response was significantly higher than the intra-individual variability (*i.e*. high and significant Fisher’s F in Supplementary File 2). In addition, compared to those with silenced KCs, inter-animal variability in animals with a functional MB contributed to a higher fraction of the overall variance in odor response (*i.e.* compare the r2 coefficients in Supplementary File 2). The MB could therefore be a significant source of inter-individual variability in odor-evoked responses of CN-33.”

Discussion

“We observed high variability in the responses of both CN-33 and MBON-m1 both to activation of the whole MB pathway, as well as to odor presentation. […] Therefore, the average response across untrained individuals might be similar to the response of a single individual in a naive neutral state (with any interindividual differences averaged out across a population), and the variability may represent the degree of freedom for the MB to tune this response to a stimulus depending on previous experience or state.”

Similarly, a key result in Figure 5a leads the authors to claim that MBON-m1 is modified by experience. In the main figure, they (unusually) do not plot the individual traces, which prevents appreciation of the variability evident in the corresponding supplementary figure (ED Figure 9): here, out of nine animals imaged, only three show dramatically diminished odor responses post-conditioning (in several cases there is little odor response even pretraining; and the non-conditioned odor can, in some cases, also show diminished responses). None of the nuance is really captured in what the authors state: "As expected, we found significantly decreased responses of MBON-m1 to the paired (CS+) and not the unpaired odor".

Even when using the classical, manual training paradigms in freely behaving animals not all individuals learn. Thus, in a training paradigm in freely behaving animals (pairing of odor with optogenetic Basin activation) the mean learning score across multiple batches of animals was -0.3 (see Figure 1e in Eschbach et al., 2020. Nature neuroscience) and the mean preference index for the aversively conditioned odor was also -0.3 (N = 20 batches; see Supplementary Figure 2b in Eschbach et al., 2020. Nature neuroscience).

The learning performance score is defined as [preference index (paired) – preference index (unpaired)]/2. If all animals from the paired group were avoiding an aversively conditioned odor, and all animals in the unpaired group were approaching it, then the learning score would be close to -1. Thus, an average learning score of -0.3 indicates that while, on average, there is a significant difference between paired and unpaired groups, not all animals learn to avoid the conditioned odor. This score means that, on average, the number of larvae that approach the conditioned odor is decreased by 30%, i.e. that ca. ⅓ of animals have learnt to avoid the odor.

The preference index for an odor is defined as [N(larvae close to odor source) – N(larvae far from odor source)]/N (total). The average preference score for the aversively conditioned odor in the paired group was -0.3, indicating that, on average, 30% more larvae are found far from the odor source than close to it. Also, an appetitive associative memory has been shown to be formed to the unpaired odor in larvae undergoing an aversive odor training (Schleyer et al., 2018), which could be consistent with an increased response of MBON-m1 to the learned odor.

In the present study we found MBON-m1 has dramatically diminished responses to CS+ in 3/9 animals and mildly reduced responses to CS+ in a further 4/9 animals (Figure 5a). MBON-m1 also had dramatically increased response to CS- in 3/9 animals (Figure 5a). These findings are largely consistent with the learning performance obtained by training freely behaving animals. We would like to note that this is the first time in the larval field that individual immobilised animals were trained in the microscope and it would have been possible that restrained animals did not learn as well as unrestrained ones. The finding that ⅓ of restrained animals showed dramatically diminished MBON-m1 responses to CS+, suggests they learn with similar probability to freely crawling animals.

It is possible that with different training conditions, such as more training trials, longer overlap between CS and US or higher intensity of Basin activation, the probability of learning (and depressing the MBON response) could increase. However, identifying the conditions that give higher learning probabilities and stronger reductions in MBON response to CS+, is a long-term effort that requires systematic variation of one parameter at a time. Such a project is beyond the scope of this study.

We would also like to point out that compared to training populations of freely behaving animals, loading animals into a microfluidic device and training them in a two-photon microscope is extremely time consuming and low-throughput. Thus, it is not feasible to obtain high Ns in such experiments.

We have shown that we do have a statistically significant reduction in MBON-m1 response in paired group compared to unpaired group. This suggests that MBON-m1 response to conditioned odor is reduced through aversive training.

To address the Reviewers’ point we have therefore modified Figure 5a to show traces from individual animals, and not just the average response. We have also modified the text in the Results section to stress the variability in MBON-depression across individuals, and to mention that this level of variability is consistent with other training paradigms in freely behaving animals:

“We found that the response to conditioned odor was variable across individuals (Figure 4 —figure supplement 2d), but on average, we found significantly decreased responses of MBON-m1 to the paired (CS+) and not the unpaired odor (CS-, Figure 5a, Supplementary File 2). In some individuals we also observed an increased response to the CS- after training (Figure 5a, Figure 4 —figure supplement 2d). In previous studies temporally pairing Basin activation with an odor in freely behaving animals induced an aversive memory where the mean learning score across multiple batches of animals was -0.3 (*i.e.* on average, the number of larvae approaching the odour was decreased by 30%) [36]. Also, an appetitive associative memory has been shown to be formed to the unpaired odor in larvae undergoing aversive odor training [69]. Thus, our results after training immobilised animals in the microscope, showing a dramatic decrease in MBON-m1 response to CS+ in 3 out of 9 animals, and an increased response to CS- in another 3 animals (Figure 5a), are by and large consistent with learning scores in freely behaving animals.”

I have similar concerns with the physiological analysis of the other investigated neuron population (CN-33) (Figure 6c-d).

To address this point we have provided additional experimental evidence for the existence of indirect functionally excitatory and inhibitory connections from KCs onto CN-33, consistent with the prediction from the connectome and added new Figure 6d. We activated KCs and recorded CN-33 responses in the absence and presence of GABA (picrotoxin, PTX) or AChR blockers (mecamylamine, MCA). A strong excitatory response to KC appeared when PTX was added to the solution and disappeared when MCA was added to the solution. These results demonstrate that CN-33 receives both functional inhibitory and excitatory inputs from the MB pathway, in accordance with the anatomical data that shows it receives synaptic input from GABAergic, glutamatergic, and cholinergic MBONs.

We have added a new Figure 6d showing the responses of CN-33 before and after perfusion with GABAR and AchR blockers.

We have added the following section in the Results:

“CN-33 receives direct synaptic input both from inhibitory avoidance-promoting and from excitatory approach-promoting MBONs, which could counterbalance each other. Consistent with this anatomical connectivity, adding the GABA/vGluT receptor blocker, PTX, revealed strong excitatory responses of CN33 to KC activation. […] As for MBON-m1, we found high interindividual variability of response to KCs activation compared to intraindividual variability (*i.e.* significant Fisher’s F, Supplementary File 2), which suggests that the balance of excitatory vs. inhibitory drive from MB pathways onto CN-33 is individual-specific.”

For this populations, they do not examine the impact of learning (even though in the abstract they may a rather general statement about convergence neurons: “Aversive learning skews their inputs towards inhibition to modulate turning).”

To address this comment we have significantly modified our claims in the abstract and text to focus on the structural and functional connections from MB and LH onto CN-33:

Abstract:

“Animal behavior is shaped both by evolution and by individual experience. Parallel brain pathways encode innate and learned valences of cues, but the way in which they are integrated during action-selection is not well understood. We used electron microscopy to comprehensively map with synaptic resolution all neurons downstream of all Mushroom Body output neurons (encoding learned valences) and characterized their patterns of interaction with Lateral Horn neurons (encoding innate valences) in *Drosophila* larva. The connectome revealed multiple *convergence neuron* types that receive convergent Mushroom Body and Lateral Horn inputs. A subset of these receives excitatory input from positive-valence MB and LH pathways and inhibitory input from negative-valence MB pathways. We confirmed functional connectivity from LH and MB pathways and behavioral roles of two of these neurons. These neurons encode integrated odor value and bidirectionally regulate turning. Based on this we speculate that learning could potentially skew the balance of excitation and inhibition onto these neurons and thereby modulate turning. Together, our study provides insights into the circuits that integrate learned and innate to modify behavior.”

Based on the structural and functional connections, in the Discussion we speculate that aversive learning could skew CN-33 responses to learning in a similar way to MBON-m1:

Discussion:

“One subtype of 10 convergence neurons (CNs) receives excitatory input from positive-valence MB and LH pathways and inhibitory input from negative-valence MB pathways. We confirmed functional connectivity from LH and MB pathways and behavioral roles of two of these neurons. These neurons encode integrated odor value (coding for positive value with an increase in their activity) and regulate turning. They are activated by an attractive odor, and when activated they repress turning. Conversely, when inactivated, they increase turning. Based on this we speculate that learning could potentially skew the balance of excitation and inhibition onto these neurons and thereby modulate turning. For one of these neurons we indeed verified that aversive learning skews inputs towards inhibition. Together, our study provides insights into the circuit mechanisms by which learned valences could interact with innate ones to modify behavior.”

**Author response table 1. sa2table1:** 

Detailed statistics (Analysis of variance):					
'Source'	'Sum Sq.'	'd.f.'	'Mean Sq.'	'F'	'Prob>F'
'indiv'	0.26	4	0.065	10.9	2.8e-07
'CS'	0.09	2	0.046	7.8	0.0007
'indiv*CS'	0.11	8	0.014	2.4	0.021
'Error'	0.55	93	0.006	[]	[]
'Total'	1.00	107	[]	[]	[]

In sum, for such complex experiments, and complex network interactions, a much larger dataset would be needed to solidify the authors’ claims. Without further experimental data, the interpretation should be significantly toned down.

To address these comments we have added new functional connectivity and calcium imaging experiments and we have significantly toned down some of the interpretations.

i. We have performed new functional connectivity experiments (activating KCs and imaging in MBONm1 and CN-33) before and after perfusion with GABAR and AchR blockers. These experiments demonstrate that MBON-m1 and CN-33 receive indirect functional excitatory and inhibitory input from KCs, consistent with the predictions from the connectome. We have added new Figure 4e and 6d with these results.

ii. We have performed a low N of new experiments in which we imaged CN-33 responses to conditioned and unconditioned odor following aversive training. These experiments show that CN-33 response to conditioned odor is significantly reduced. We have included Figure for Reviewers with these results.

iii. Since we could not generate a large N for the above learning experiments due to the pandemic and lack of childcare, we have removed the speculation about the role of CN-33 in learning from the Abstract and the Results to focus only the structural and functional connections from MB and LH onto CN-33. Based on the structural and functional connections, in the Discussion we speculate that aversive learning could skew CN-33 responses to learning in a similar way to MBON-m1.

In summary, the main findings and contributions of our manuscript are:

1. Providing, for the first time, a comprehensive synaptic-resolution wiring diagram of the circuitry downstream of mushroom body output neurons (MBONs) and lateral horn neurons (LHNs) in *Drosophila* larva.

2. Identifying, for the first time, the actions evoked by the activation of individual larval MBON types. We show that MBONs from aversive memory compartments promote approach (promote crawling and repress turning), whereas MBONs from appetitive compartments promote avoidance (promote turning and repress crawling).

3. Integrating the structural information with neurotransmitter profiles of MBONs and their roles in promoting actions.

4. Comprehensive characterisation of the patterns of convergence between MBONs and LHNs in *Drosophila* larvae.

5. Discovery of CNs that receive convergent inputs of opposite sign and opposite valence from MB and LH pathways.

6. For two members of this population we demonstrate they receive functionally excitatory connections from an attractive LH pathway and both functionally excitatory and inhibitory connections from the MB pathway. This functional connectivity is consistent with the structural input from both excitatory and inhibitory MBONs. Interestingly, the excitatory MBONs that provide input to these two CNs are approach-promoting, and the MBONs that provide inhibitory input to these CNs are avoidance-promoting.

7. For these two members of the CN population we also demonstrate their activation promotes approach and their inhibition promotes avoidance, consistent with the findings that they receive excitatory input from MBONs and LHNs that promote approach and inhibitory input from MBONs that promote avoidance.

8. For one of these CNs (MBON-m1) we also show that aversive training reduces their responses to paired odor, compared to unpaired odor.

9. Based on these findings we speculate in the Discussion that the newly discovered population of CNs could compute integrated odor value (by comparing their excitatory approach-promoting and inhibitory avoidance-promoting inputs) and promote actions based on this integrated value. While it is beyond the scope of this already large study to investigate in detail the role of each of these CNs in memory-based action selection, we think that identifying a candidate population of neurons that could be involved in the computation of integrated value and action selection is very important and will inspire numerous targeted follow-up studies.

Just the EM reconstruction alone is a valuable resource for the community and would merit publication on its own.

We have gone much further to test functionally some of the structural connections observed in the connectome and to test the behaviors evoked by MBONs and some CNs. We have further confirmed that conditioned odor response of at least one of the CNs (MBON-m1) is modified by associative learning. Based on these findings we speculate in the Discussion that other newly discovered populations of CNs could be modified in a similar way.

We believe that the structural and functional connectivity findings and behavioral activation results merit publication because they provide a slew of new circuit motifs and candidate pathways that are a valuable resource for many future follow up studies.

Reviewer #2:This manuscript uses a combination of EM-based connectomics, behavior, and calcium imaging to explore how learned and innate valence signals are integrated to drive action selection in *Drosophila*. Building on previous work where the authors characterized the valence of memory encoding for mushroom body compartments, they then characterize the valence encoding of the output neurons for each of these compartments, and reconstruct all downstream targets. They find a complex network of downstream neurons which receive input from mushroom body output neurons encoding opposite valances. Using behavior, optogenetics, and calcium imaging the authors explore how two such convergence neurons might integrate both learned and innate valence, as well as signals of opposite valence.This work exploring the integration of valence signals in the *Drosophila* olfactory system represents an interesting parallel with the author’s other recent work exploring how memory encoding in the mushroom body. Most of the conclusions are well supported. However, there are a number of issues that should be addressed prior to publication.Major Concerns:1. Conclusions of the papera. The primary hypothesis of this paper seems to be that in addition to depression of the KC-MBON connection due to aversive learning, shown in (Eschbach 2020), there exist other mechanisms for the comparing of learned valences, including integration at both the MBON and CN level.

We are not sure what the Reviewer means by “shown in Eschbach 2020”. We had not shown that MBONs are depressed by learning in Eschbach 2020, although studies in adult *Drosophila* have shown this. Here, we have shown that MBON-m1 response to CS+ is depressed by learning (Figure 5a). Our main question in this study was about the way in which inputs from distinct MBONs (encoding distinct valences) are integrated with each other and with LH inputs.

One proposed mechanism is that MBONs encoding opposite valences inhibit one another. The connectome only somewhat supports this mechanism. Of the tested MBONs, only 2 show the strict lateral inhibition described. Three have no interactions, two inhibit MBONs with both the same and opposite valence, one excites MBONs of both the same and opposite valence, and one inhibits an MBON of the same valence. Based on these connection motifs, can it really be said that lateral inhibition is a general principle for mushroom body organization?

We agree with this point. Our intention was not to suggest that lateral inhibition was the main mechanism for comparing learned valences. In fact, lateral inhibition between MBONs could only occur for some compartments (tip of the VL, CA, upper toe of the ML, LA and MBON-m1), but not for others. On its own it would not be sufficient to explain a winner-take-all action selection mechanism. Our intention was to propose that lateral inhibition could contribute to comparing learned valences and increasing the contrast between them. We would argue, however, that the existence of lateral inhibition between at least some MBONs of opposite valence is one of the general principles of the *Drosophila* MB organisation, since it has been found both in the adult (Felsenberg *et al.*, Nature, 2017) and in the larva (in this study).

We have modified the relevant section in the Results in the text to clarify these points:

“We therefore combined the behavioral effects of MBON activation with this information (Figure 2i). […] We therefore investigated if interactions between MBONs could be found one synapse downstream of the MBONs.”

b. The latter half of the paper focuses on two CNs, one an MBON itself, and another a feedback neuron (described in Eschbach 2020). The author’s primary argument is that these neurons integrate both learned and innate valences, as well as are integrating inputs of MBONs of opposite valence.

Our connectome has revealed that CN-33 and MBON-m1 receive strong (more than 10 synapses) and reliable (in both hemispheres) direct synaptic inputs from excitatory and inhibitory MBONs of opposite valence. Thus, MBON-m1 receives GABAergic and glutamatergic input from avoidance-promoting MBON-i1 and -h1, and cholinergic input from approach-promoting MBON-e1. CN-33 receives glutamatergic input from avoidance-promoting MBON-i1 and MBON-e2, and cholinergic input from the approach-promoting MBON-e1.

However, we do agree that we had not previously shown that the structural inputs from the cholinergic, GABA-ergic and glutamatergic MBONs are functional.

To address this point we have therefore designed experiments to test whether CN-33 and MBON-m1 receive convergent functional inhibitory and excitatory input from their presynaptic MBONs, consistent with the structural connectome. We do not have LexA lines for individual MBONs (or CNs) to activate them directly. Instead, we activated KCs to indirectly activate MBONs and imaged the responses of CN33 and MBON-m1 to pan-KC activation before and after perfusion with GABAR/vGluR (picrotoxin, PTX) and AchR (mecamylamine, ME) blockers. We observed significantly increased excitatory responses to KC activation in the CNs after perfusion with the GABAR/vGluR blocker, demonstrating functional excitatory inputs from the MB pathway, consistent with the structural inputs onto these neurons from cholinergic MBONs. This result also demonstrates that MBON-m1 and CN-33 receive strong inhibitory (GABAergic or glutamataregic) inputs which can prevent the excitatory response, consistent with structural inputs from GABAergic and glutamatergic MBONs. The excitatory responses were abolished in the presence of AchR blocker. In addition, when bathed with saline, the CNs show KC-evoked mixed responses (excitatory, inhibitory, or absent), which is visible when considering the overall absolute amplitude in fluorescence change. This result is consistent with the prior findings in vivo that we reported in the manuscript in Figure 4d and 6c.

In our view these new functional connectivity experiments in the presence and absence of blockers provide conclusive evidence for the existence of both functionally excitatory and inhibitory connections from the MB onto CN-33 and MBON-m1. These functional inputs are consistent with the direct synaptic inputs from cholinergic and GABAergic or glutamatergic MBONs onto CN-33 and MBON-m1. The functionally excitatory connection from KCs onto MBON-m1 could be mediated both by KCs and excitatory MBONs. However, KCs are not known to be inhibitory, so the functionally inhibitory connections from KCs onto MBON-m1 must be indirect, mediated by MBONs. CN-33, does not receive any direct KC input, so both the functional excitatory and inhibitory connections must be mediated by the MBONs. At the same time the connectome shows that CN-33 receives inhibitory input from MBON-i1 and -e2 that promote avoidance and excitatory input from MBON-e1 that promotes approach. Similarly, MBON-m1 receives inhibitory input from MBON-i1 and- h1 that promote avoidance and cholinergic input from MBON-e1 that promotes approach. Thus the direct inhibitory inputs onto CNs are provided by MBONs that promote avoidance, and direct excitatory inputs are provided by MBONs that promote approach. Together, this structural and functional connectivity data provides strong support for the idea that CN-33 and MBON-m1 integrate input of opposite sign from MBONs of opposite valence. However, we cannot also exclude the possibility that some of the functionally excitatory and inhibitory input we observed in our functional connectivity experiments is mediated by MB2Ons that are downstream of these MBONs and that also synapse onto CNs.

We have added new Figures 4e and 6d with these results and new sessions in the Results sections. We have also mentioned that some of the functional inputs may be mediated by MB2Ons.

“Our observation that MBON-m1 has excitatory response to KC activation in some individuals, inhibitory response in some, or no response in others suggests it receives both excitatory and inhibitory inputs from the MB pathway. […] We speculate that learning could modulate the balance of excitation and inhibition onto MBON-m1 by modifying the strength of direct KC input onto MBON-m1 as well as KC input onto excitatory or inhibitory MBONs that synapse onto MBON-m1.”

“CN-33 receives direct synaptic input both from inhibitory avoidance-promoting and from excitatory approach-promoting MBONs, which could counterbalance each other. Consistent with this anatomical connectivity, adding the GABA/vGluT receptor blocker, PTX, revealed strong excitatory responses of CN33 to KC activation (Wilcoxon test, p<0.01). This experiment confirms that CN-33 receives functional inhibitory and excitatory input from KCs (Figure 6d). Since CN-33 does not receive any direct KC input but does receive direct MBON input, these functional connections must be indirect and mediated by the MBONs (and possibly also by MB2Ons that receive input from these MBONs and synapse onto CN-33).”

The data presented is consistent with the idea that both MBON-m1 and CN-33 integrate information from both the LH (innate) and the mushroom body (learned), however, it is not entirely clear that there is sufficient functional evidence to support the idea that either of these neurons is integrating MBON input of opposite valence. The connectome does suggest that this is the case, however, silencing of the mushroom body had no effect on the response of either MBON-m1 or CN-33 to odors. Pairing the odor with an aversive stimulus clearly demonstrates that the neurons are capable of integrating innate and learned stimuli, but it is unclear if the inputs from positive MBONs are being depressed as shown in the schematic in Figure 7.2. Behavioral experimentsa. The optogenetic stimuli used for conditioning experiments in Figure 2 produce a large (startle?) response, on top of which all other behavior is measured. What evidence is there that the behavioral conditioning experiments in Figure 2 are not producing a potentiation or reduction of this light-induced startle response? This is particularly worrisome in the majority of cases where the dynamics of the conditioned behavior follow a similar time-course to the startle response. Overall, it is not convincing that this analysis is cleanly separating MBONs that would drive avoidance and approach independent of the light artifact.

There is indeed a consistent turning startle response to red light exposure, which appears potentiated or abolished when MBONs are activated at the same time. Our hypothesis is that this apparent modulation of the startle response indicates a role of the MBONs in modulating turning (in response to odor), in parallel and independently of the modulation of turning by the light pathway, with the two kinds of inputs (MBON activation and light) likely linearly summed for the expression of turning behavior. This reasoning is based on the reconstruction of the visual pathway which appears to converge with the odor pathway directly downstream of the LH (*i.e.* at the same level as the CN layer, with some overlap), and on the findings by Gepner *et al.,* (*eLife*, 2015) which showed that light and odor pathways interact linearly on turning behavior. However, we agree that based on the experiments presented in Figure 2 it was not possible to exclude that MBONs are not sufficient to modulate turning on their own, but they can only modulate the light-evoked turning responses.

To address this point we therefore performed another series of behavioural experiments where larvae were constantly exposed to blue light of low intensity and red light with randomly varying intensity (a white noise stimulus). Under these conditions control larvae adapt to the light stimulus and ignore the changes in variation of red light stimuli, and do not show red light-induced turning responses (See empty Split control in new Figure 2 —figure supplement 3a-c). However, we found that the changes in red light intensity still modulated turning in individuals expressing CsChrimson in specific MBONs, demonstrating that MBONs can promote or repress turning and crawling even when there is no response to light. Thus, their effect is independent of the light-induced startle response.

In these experiments we performed reverse-correlation analysis to further demonstrate the role of MBONs in promoting approach or avoidance, as previously done for ORs (Gepner *et al.*, *eLife* 2015) and asked whether the light intensity (*i.e.* the activity of MBON was reducing or increasing prior to turns).

We tested a subset of MBONs whose activation promoted turning in our previous experiments (avoidance-promoting) and a subset whose activation repressed turning (approach-promoting).

Consistent with our previous results we found that activity of the two sets of MBONs had opposite correlation with turning behaviour (new Figure 2 —figure supplement 3a-b).

For the avoidance-promoting MBONs from appetitive compartments, turns occurred after a significant increase in light intensity (*i.e.* after an increase in their activity). This is entirely consistent with our prior finding that activating these MBONs increased turning (at the onset of light).

For the approach-promoting MBONs from aversive compartments turns occurred after a significant decrease in light intensity (*i.e.* after a decrease in their activity). This is entirely consistent with our prior finding that reducing the activity in these MBONs increased turning (at the offset of light).

These results are further consistent with the idea that an increase in the activity of avoidance promoting MBONs promotes turning, resulting in avoidance of an odor whose concentration is increasing. Conversely, a decrease in the activity of approach-promoting MBONs promotes turning. This ensures that if the animal is crawling away from an attractive odor source it turns to correct its course.

These experiments also enabled us to analyse how the increase or decrease in MBON activity during a single head cast (turn) influences the likelihood of acceptance or rejection of that turn direction (new Figure 2 —figure supplement 3c). Larvae are known to accept turns during which they sense an increase in the concentration of an attractive odor and reject casts during which they sense an increase in the concentration of an aversive odor (Gomez-Marin and Louis 2012, Gershow et al., 2012).

The reverse correlation analysis showed that, for avoidance-promoting MBONs, the greater the light intensity (*i.e.* MBON activity) during a head cast, the greater the likelihood of rejecting that direction and performing another head-cast. Conversely, for approach-promoting MBONs, the greater the decrease in light intensity (*i.e.* MBON activity) during a turn, the higher the likelihood of accepting that direction (and not performing another head cast).

We believe these new experiments clearly demonstrate that MBON activity directly affects actions involved in odor navigation, such as the probability of turning and the probability of accepting a new direction after a head-sweep.

We have added new Figure 2 —figure supplement 3a-c. with these results and the following section in the Results:

“In our experiments we observed a turn response to the red light used for optogenetic activation in control animals. We wanted to rule out the possibility that MBONs only modulate the light-evoke startle and confirm they could modulate turning even in the absence of a light-evoked turn. For a subset of positive- and negative-valence MBONs we therefore repeated the optogenetic manipulation experiments in a different way, by constantly exposing larvae to dim blue light of constant intensity and bright red light (for CsChrimson activation) of randomly varying intensity (Figure 2 —figure supplement 3a-c). Under these conditions, control larvae habituated to light and did not show light-induced startle responses. Reverse-correlation analysis[51,56,57] between turning and red light intensity revealed that larvae with CsChrimson in negative-valence MBONs turn following an increase in red light intensity (*i.e.* an increase in MBON activity), and larvae with CsChrimson in positive-valence MBONs turn following a decrease in light intensity (*i.e.* a decrease in MBON activity, Figure 2 —figure supplement 3a-b). Also, for negative-valence MBONs, the greater the change in light intensity during a turn, the higher the probability of rejecting that direction and performing another turn. Conversely, for positive-valence MBONs, the greater the change in light intensity during a turn, the higher the probability of accepting that turn direction (Figure 2 —figure supplement 3a and 3c). This indicates that activity in MBONs impacts multiple behavioral components involved in avoidance or approach of a sensory stimulus in a way that is consistent with their valence.”

b. Each MBON in Figure 2 is classified as driving either approach or avoidance, but the dynamics of their angular steering and speed vary widely. This should at least be discussed in the text, and the authors should justify their decision for categorization in light of this variability.

This is an interesting point. In the text, we had discussed the existence or absence of a turning response to the reduction in activation (at the offset of light) for some MBONs, but we did not indeed mention other kinds of variability.

Thus, while all avoidance promoting MBONs increase turning in response to an increase in their activity, relative to controls, some of them evoke stronger turning responses than others, and some of them evoke prolonged turning responses compared to others. For example MBON-a2 promotes turning throughout the period in which it is activated, whether some adaptation of the turning response is observed for other MBONs.

Similarly, while all avoidance-promoting MBONs promote turning in response to an increase in their activity, only two of them also promote crawling in response to a decrease in their activity. Thus, the reduction in MBON-h1/h2 and MBON-i1 evokes an increase in crawling speed compared to controls. This response to light-offset is not observed for MBON-a2, -k1, and e2.

Conversely, while all approach-promoting MBONs repress turning in response to an increase in their activity, two of them (MBON-m1 and MBON-a1) also evoke an increase in mean crawling speed, compared to controls, while others do not.

Finally, three approach-promoting MBONs evoke an increase in turning, compared to controls, in response to a decrease in their activity at light offset (MBON-a1, -g1/g2, and m1), but others do not.

This does indeed suggest that even MBONs of the same type could provide slightly different contributions to the avoidance or approach response. This could potentially enable the larva to learn specific types of avoidance or approach responses depending on the type of punishment or reward.

We have now added a description of the variable effects of different MBONs and a speculation about the possible roles of this variability in learning specific conditioned responses:

“Activating some MBONs repressed turning and promoted crawling, compared to controls (Figure 2bc and e-g). […] This suggests that the MB could potentially allow different types of aversive memories to be differentially expressed (with varying durations of turns, speeds of crawling or efficacies of navigational strategies).”

3. Calcium Imaging experimentsThe calcium imaging data throughout the paper is quite noisy and responses are variable across individual animals. The authors sometimes attribute this variability to biological variability in circuit connectivity (eg, Figure 6). Where this is done, it is worth including an explicit comparison of within fly vs across variability, to test the degree to which responses are stereotyped within an animal.

In response to this comment we have now applied ANOVA on the data in order to assess how stereotyped the responses are within animals (see new Supplementary File 2).

Of note, the variability of responses to the optogenetic activation of KCs, or to natural odor is primarily inter-individual rather than intra-individual: the Anova analysis shows a strong individual effect on the response both for MBON-m1 and CN-33 (new Supplementary File 2).

Furthermore, we also found that the inter-animal variability in response to odor was larger (relative to the overall variance) when the MBs are functional, than when MBs are inactivated (in animals in which KCs express TNT; new Supplementary File 2). This suggests that the inter-animal variability is coming from the learning pathway, rather than from the innate pathway.

We have added new sections in the Results and Discussion mentioning this analysis.

Results

**“**Interestingly, odor-evoked responses were highly variable across individuals (ranging from inhibition to excitation), even though, on average across individuals the response was excitatory (Figure 4c). ANOVA revealed that inter-individual variability was significantly higher than the intra-individual variability (*i.e.* high and significant Fisher’s F in Supplementary File 2). Compared to those with silenced KCs, inter-animal variability in animals with a functional MB contributed to a higher fraction of the overall variance in odor response (Figure 4c, and compare the r^2^ coefficients in Supplementary File 2). This suggests that the MB could be a significant source of inter-individual variability in odor-evoked responses in MBONm1.”

**“**We found that the inter-individual variability of MBON-m1 responses to KC activation was significantly higher than the intra-individual variability (see Fisher’s F in Supplementary File 2). The variability in the response to KC activation between individuals could be due to a number of factors, including slightly different levels of CsChrimson expression, different states or different experiences.”

“As was the case for MBON-m1, ANOVA revealed that inter-individual variability in CN-33 odor response was significantly higher than the intra-individual variability (*i.e*. high and significant Fisher’s F in Supplementary File 2). In addition, compared to those with silenced KCs, inter-animal variability in animals with a functional MB contributed to a higher fraction of the overall variance in odor response (*i.e.* compare the r^2^ coefficients in Supplementary File 2). The MB could therefore be a significant source of inter-individual variability in odor-evoked responses of CN-33.”

**“**As for MBON-m1, we found high interindividual variability of response to KCs activation compared to intraindividual variability (*i.e.* significant Fisher’s F, Supplementary File 2), which suggests that the balance of excitatory vs. inhibitory drive from MB pathways onto CN-33 is individual-specific.”

Discussion:

**“**We observed high variability in the responses of both CN-33 and MBON-m1 both to activation of the whole MB pathway, as well as to odor presentation. […] Therefore, the average response across untrained individuals might be similar to the response of a single individual in a naive neutral state (with any interindividual differences averaged out across a population), and the variability may represent the degree of freedom for the MB to tune this response to a stimulus depending on previous experience or state.”